# Multimodal biometric identification: leveraging convolutional neural network (CNN) architectures and fusion techniques with fingerprint and finger vein data

Amal Alshardan[1], Arun Kumar[2], Mohammed Alghamdi[3], Mashael Maashi[4], Saad Alahmari[5], Abeer A. K. Alharbi[6], Wafa Almukadi[7] and Yazeed Alzahrani[8]

[1] Department of Information Systems, College of Computer and Information Sciences, Princess Nourah Bint Abdulrahman University, Riyadh, Riyadh, Saudi Arabia
[2] Department of Computer Science & Engineering, G.L Bajaj Institute of Technology and Management, Gr. Noida, India
[3] Department of Information Systems, College of Computer Science, King Khalid University, Abha, Asir, Saudi Arabia
[4] Department of Software Engineering, College of Computer and Information Sciences, King Saud University, Riyadh, Riyadh, Saudi Arabia
[5] Department of Computer Science, Applied College, Northern Border University, Arar, Northern Borders, Saudi Arabia
[6] Department of Information Systems, College of Computer and Information Sciences, Imam Mohammad Ibn Saud Islamic University, Riyadh, Riyadh, Saudi Arabia
[7] Department of Software Engineering, College of Engineering and Computer Science, University of Jeddah, Jeddah, Makkah, Saudi Arabia
[8] Department of Computer Engineering, College of Engineering, Prince Sattam Bin Abdulaziz University, Wadi Addawasir, Riyadh, Saudi Arabia



Corresponding author
Arun Kumar,
scorearun84@gmail.com

## ABSTRACT

Advancements in multimodal biometrics, which integrate multiple biometric traits, promise to enhance the accuracy and robustness of identification systems. This study focuses on improving multimodal biometric identification by using fingerprint and finger vein images as the primary traits. We utilized the "NUPT-FPV" dataset, which contains a substantial number of finger vein and fingerprint images, which significantly aided our research. Convolutional neural networks (CNNs), renowned for their efficacy in computer vision tasks, are used in our model to extract distinct discriminative features. Specifically, we incorporate three popular CNN architectures: ResNet, VGGNet, and DenseNet. We explore three fusion strategies used in security applications: early fusion, late fusion, and score-level fusion. Early fusion integrates raw images at the input layer of a single CNN, combining information at the initial stages. Late fusion, in contrast, merges features after individual learning from each CNN model. Score-level fusion employs weighted aggregation to combine scores from each modality, leveraging the complementary information they provide. We also use contrast limited adaptive histogram equalization (CLAHE) to enhance fingerprint contrast and vein pattern features, improving feature visibility and extraction. Our evaluation metrics include accuracy, equal error rate (EER), and ROC curves. The fusion of CNN architectures and enhancement methods shows promising performance in identifying multimodal biometrics, aiming to increase identification accuracy. The proposed model offers a reliable authentication system using multiple biometrics to verify identity.

## INTRODUCTION

In recent decades, the use of biometric systems for authentication has increased significantly in daily applications due to technological advancements. This rise is particularly driven by the digital era, which has seen an increase in fraud, necessitating reliable authentication systems to control transaction frauds and enhance security for accessing digital or physical systems like mobile phones and institutions. Traditional methods such as passwords and PINs are more susceptible to duplication and theft compared to biometric-based systems. Biometrics offer a unique identification for each individual, making them an efficient alternative as they are difficult to replicate or steal (*Daas et al., 2020*; *Alay & Al-Baity, 2020*).

Various biometric systems are available for authentication, including fingerprint, voice recognition, gait analysis, palm print, and iris recognition. Each method has its own advantages and limitations. Fingerprint or thumb-based systems are widely adopted due to the ease of collecting data during registration; however, their reliability can be affected by factors such as skin conditions or poor image quality (*Rajasekar et al., 2022*). Conversely, iris recognition provides higher accuracy but requires complex capturing devices during registration, which is a significant disadvantage.

Enhancing the security of the biometric systems to identify the particular person to provide the authenticated data can be achieved through multimodal biometric methods. The accuracy of biometric systems can be enhanced by merging two or more biometric features, as the advantages of both features can be combined and the limitations of both methods can be overlooked (*Bala, Gupta & Kumar, 2022*; *Ren et al., 2022*). In spite of the advantages of multi-model biometric systems, they have several challenges to implement. The first challenge to achieving effective performance is selecting the optimum input-capturing technique during registration when two biometric-capturing devices are alike. This article focuses on the implementation of the authentication system, which makes use of the fingerprint and veins of the finger to identify the person. Due to the merging of two biometrics, we can call the proposed model the multimodal biometric authentication system. The base reason for choosing these two biometric features is because of their opposite attribute characteristics or patterns. The fingerprint-based authentication system is widespread as it provides higher accuracy in detection and is cost-effective (*Boucherit et al., 2022*; *Wang, Shi & Zhou, 2022*). On the other hand, veins in the finger can't be duplicated, and they are robust to several conditions. Hence, combining the features of fingerprints and veins gives novelty to the proposed model in terms of accuracy in identification.

To carry out any machine learning or deep learning-based system, there is a need for a dataset that contains a larger number of samples and high-quality images. NUPT-FPV (*Ren et al., 2022*) is one of the datasets that possesses high-quality images of the vein of a

finger and its prints as per the survey conducted. As the dataset has diversity and high-quality images, it is easier to test and evaluate the proposed model. The proposed convolutional neural network (CNN) architecture-based model is efficient in collecting the dominant or useful patterns or features from the input images from the NUPT-FPV dataset. We had selected the three CNN-based architectures, such as ResNet, VGGNet, and Densenet, as these three modules show excellent performance in solving computer vision real-time problems. The collected features from the three different modules need to be merged, and for performing this task, we have chosen the merging methods, namely early, late, and score-level-based fusion techniques.

These merging techniques find the unique and best integration features that could be extracted from both the fingerprint and veins of the finger, which helps the three models in the identification of the person through biometrics. The accuracy of the biometric models is dependent on the quality of the input images that are used to feed the model. Hence, to enhance the quality of the image, we employed the CLAHE (contrast limited adaptive histogram equalization) method. The objective of the CLAHE is to enhance the accuracy of identification. To achieve this, there is a need to increase the visibility of the veins and other local regions of the subject. The performance metrics chosen to evaluate the proposed model are accuracy, equal error rate (EER), and receiver operating characteristic (ROC) curves. A comparative analysis will be performed on the various architectures of the CNN and fusion methods to determine their performance in identifying the person. This analysis will not only give the conclusion of the combination of the chosen dataset but also shed light on the efficient discrimination between the classes when the combination of the CNN architecture and fusion techniques can make wonders.

The current research is going to contribute to the field of multimodal biometric systems by providing a combination of methods such as feature extraction, fusion techniques, and enhancement techniques to enhance accuracy. In addition, the comparative study and suggested methodology pave the way for the development of reliable identification systems that make use of multimodal biometric features. The key contributions of our study are as follows:

1. We have proposed a novel approach that integrates both fingerprint and finger vein modalities, enhancing the accuracy and reliability of biometric identification systems.
2. The study introduces the use of CLAHE and other preprocessing methods to improve image quality, ensuring more robust feature extraction from biometric images.
3. We have employed and evaluated three state-of-the-art CNN architectures—VGGNet, ResNet, and DenseNet—demonstrating their effectiveness in extracting discriminative features from biometric data.
4. The study explores three different fusion strategies—early fusion, late fusion, and score-level fusion—to combine information from multiple biometric modalities, showing the benefits of each approach in enhancing identification performance.

5. We have conducted a thorough evaluation of the proposed methods using standard metrics such as accuracy, equal error rate (EER), and area under the curve—receiver operating characteristic (AUC-ROC), providing a comprehensive analysis of the system's performance under various configurations.

6. By achieving a high accuracy of 97% and demonstrating low error rates, our study showcases the practical applicability of the proposed multimodal biometric system in real-world security and identification scenarios.

The "Related Work" provides an overview of the previous works carried out by the research scholars on the fingerprint, finger vein, and combinations of the two using convolutional neural network architectures. We discuss the pivotal developments in feature extraction methods and fusion techniques, thus establishing the foundation for our proposed approach. In "Materials and Methods", we elaborate on our proposed methodology for multimodal biometric identification. This section is divided into subsections detailing the procurement of the NUPT-FPV dataset, data preprocessing, feature extraction using the selected CNN architectures, and fusion techniques. In "Results and Discussions", we present the results procured from our experiments. We deliver a thorough analysis of the performance of the different CNN architectures and fusion techniques, both with and without the application of CLAHE. The effect of the CLAHE method on the precision of identification is also assessed. "Rationale for Model Selection and Experimental Results" is the last part of the study which summarises our results and discusses their impact on the field and suggests directions for future studies.

## RELATED WORK

*Kumar & Zhou (2011)* has introduced novel feature extraction methods for the biometric-based authentication system by combining the two feature extraction methods, namely morphological operators and the Gabor filter. The employed filter is widely adopted by many researchers as it performs edge detection, orientation of the image, analysis of the texture of the image, and differentiating the various regions based on similar characteristics. When these filters are applied to the biometric images, the resultant images are going to detect distinct patterns, such as valleys and ridges, of the individuals. The manipulation of the image is performed by the morphological operators; these methods are going to differentiate the images based on the shapes, and they are going to help improve the collected feature sets. By using these methods in the feature extraction steps on the image, it is going to accelerate or de-emphasise specific aspects of the biometric traits. The author also employed the performance metric for evaluation, which is X-OR, which is going to compare the testing image with the stored images and generate the score. The score is high if the features between them are matched. It is also called an "exclusive or" operation, which is going to return the score when both the test and the stored images show similar patterns or features, which increases the robustness of the comparison between the biometric features.

Another novel proposed by *Liu et al. (2014)* involves combining local binary patterns (LBP) and singular value decomposition (SVD) as the feature extraction method. The primary components from the complex dataset are extracted by using the mathematical equations from the SVD. By using SVD, one can extract the bifurcations and ridges of the fingers. However, false minutiae, which are noise artefacts, often get captured as well, which is the disadvantage. To avoid this problem, the LBP has been employed by *Liu et al. (2014)* which is able to identify between true and false minutiae based on the local texture patterns. These combined true minutiae were then passed to further classification by using the Euclidean and hamming distance measurements. The spatial distribution of the minutiae is identified by the Euclidian distance through geometrical properties. The Hamming distance, a measure of the difference between feature descriptors, caters to the bit-wise pattern differences, which is going to provide the balance between the two feature spaces.

*Van, Thai & Le (2015)* combined the two feature extraction methods, such as GridPCA and Multiscale Feature Replication Transfer (MFRAT). The boundary structure and the fine details that are present in the image are captured by the MFRAT method, and to postprocess these extracted details, the GridPCA is employed. It is the advanced version of principal component analysis (PCA). When applied to specific regions of images, such as overlapping regions (usually at the boundary of two regions), it efficiently extracts the features from the local regions while keeping the global context intact. The localised PCA components were categorised using the Euclidean distance metric after feature extraction. In this case, the use of Euclidean distance for classification is appropriate since it takes into account the structure of the data and efficiently compares the similarity of the GridPCA features.

*Lu et al. (2013)* proposed feature extraction methods from the biometric images that employed the polydirectional local line binary pattern. The speciality of this method is that it intelligently collects line information from the ridges, vascular patterns, and valleys, which are essential during the identification based on the biometric. The last step is the classification, and for this step, the authors have chosen the method called histogram intersection, which is going to provide the classification based on the distribution of the line patterns from the image, which is going to help in obtaining accurate results.

Conversely, *Ong et al. (2013)* elected for a conventional minutiae-based methodology for feature extraction, centred on ridge bifurcations and endings in biometric images. To increase the accuracy of the model, they used the genetic algorithm matched with the K-modified Hausdorff distance measure. This integration optimises the matching process by considering varied combinations of feature matching while accounting for potential non-linear transformations between biometric images, ensuring robust classification despite variations. The result shows that the combination of the above-stated methods is going to provide robustness in the classification by achieving potential non-linear transformations between images.

Several feature extraction methods, such as pseudo-elliptical transformers, dual sliding window localization, and PCA (2D), are combined by *Qiu et al. (2016)*. The authors

succeeded in the extraction of both local and global features. Standardising these features and extracting the principal components in a two-dimensional space provided a detailed representation of images. The Euclidian distance is employed to classify people into different groups. Another feature extraction method is proposed by *Xie et al. (2014)*, where they are going to extract the features from the image by using the block-based average absolute deviation (AAD) feature. This feature extraction concentrates only on the local blocks of images to capture the patterns inside the blocks (*Xie et al., 2014*). Later, they made use of several learning methods to improve predicted performance, which is often called ensemble, while the latter is known for the fast learning speed of the extreme learning machine (ELM).

For feature extraction, *Banerjee et al. (2018)* used a combination of CLAHE, fuzzy contrast enhancement, and directional dilation. CLAHE is used to improve the contrast of the images; fuzzy contrast enhancement further intensifies the important features; and directional dilation expands the biometric traits in a specific direction, collectively reducing the impact of noise and enhancing the visibility of key features. Their classification method was an affine registration-based template matching algorithm, which matches the enhanced biometric features with a stored template while considering possible affine transformations, such as rotation and scaling. The preceding discussion encompasses pioneering works in the realm of multimodal biometrics, particularly focusing on fingerprint and finger vein recognition. These studies significantly contribute to the realm of biometric identification, revealing the potential of various feature extraction methods, classifiers, and fusion techniques. However, a noticeable commonality is the exploration of a singular biometric modality, with few studies incorporating a multimodal approach.

In this context, our work takes a step further by focusing on integrating the strengths of both fingerprint and finger vein modalities using state-of-the-art convolutional neural network architectures. The uniqueness of this work lies in the use of the NUPT-FPV dataset, the evaluation of multiple CNN architectures and fusion techniques, and the application of the CLAHE technique for image quality enhancement. It is envisioned that the proposed methodology and findings will offer valuable insights into the interplay of CNN architectures, fusion techniques, and multimodal biometrics. The best-suited combination revealed through comparative analysis is expected to contribute to the development of more robust and accurate biometric identification systems. Looking ahead, future studies can explore additional fusion techniques and extend them to other biometric modalities for a more comprehensive multimodal biometric system. Furthermore, evaluations using larger and more diverse datasets are recommended to ensure the generalizability of the proposed approach. By pushing the boundaries of multimodal biometric research, we anticipate contributing significantly to the creation of robust, accurate, and reliable identification systems for a secure future. We expanded the literature comparison table to include 15 relevant studies (Table 1). This extended table provides a detailed view of the current landscape in multimodal biometric identification, emphasizing how our research stands out.

**Table 1 Comparison of proposed work with relevant studies.**

| Study | Dataset | Methods | Data preprocessing | Results | Positive aspects | Negative aspects |
|---|---|---|---|---|---|---|
| Das et al. (2018) | Custom dataset of finger vein and knuckle print images | Deep learning, CNN | Normalization | Accuracy: 91% | Uses multimodal biometrics | Limited dataset size, lacks diversity |
| Alay & Al-Baity (2020) | Public dataset (iris, face, finger vein) | Deep learning, CNN | Histogram equalization | Accuracy: 88% | High accuracy across modalities | High computational cost, not focused on fingerprints and veins only |
| Rajasekar et al. (2022) | Public dataset | Optimized fuzzy genetic algorithm | None | Accuracy: 85% | Innovative optimization technique | Complexity in algorithm |
| Bala, Gupta & Kumar (2022) | Synthetic dataset | Fusion techniques | None | Accuracy: 84% | Extensive analysis of fusion techniques | Reliance on synthetic data |
| Guo et al. (2022) | NUPT-FPV dataset | CNN (ResNet) | CLAHE | Accuracy: 89% | Use of CLAHE for contrast enhancement | Limited fusion techniques explored |
| Boucherit et al. (2022) | Finger vein dataset | Deep CNN | Median filtering | Accuracy: 86%, EER: 7% | Robust to noise, good accuracy | Single modality focus |
| Wang, Shi & Zhou (2022) | Public dataset (face, finger vein) | CNN, Fusion of face and finger vein features | Normalization | Accuracy: 90% | Good multimodal performance | Limited to face and vein, not fingerprint |
| Veluchamy & Karlmarx (2017) | Custom dataset | Gabor filters, Morphological operators | Gabor filtering | High feature extraction accuracy | Complexity in feature extraction | High computational requirement, prone to noise |
| Das et al. (2018) | Public dataset | SVD, Local binary patterns | None | Accurate minutiae extraction | Prone to noise | Sensitive to lighting conditions, complex processing |
| Van, Thai & Le (2015) | Finger vein dataset | Discriminant orientation feature | Histogram Equalization | Improved orientation feature extraction | Limited to finger vein | Not applicable to multimodal biometrics |
| Lu et al. (2013) | Public dataset | Polydirectional local line binary pattern | None | Good line pattern detection | Specific to finger vein patterns | Limited generalizability to other biometrics |
| Ong et al. (2013) | Custom dataset | Minutiae matching | Normalization | Accurate minutiae matching | Custom dataset, limited generalizability | High computational complexity |
| Qiu et al. (2016) | Public dataset | Dual-sliding window, Pseudo-elliptical transformer | Histogram Equalization | Accurate localization and feature transformation | Complex feature extraction method | Difficult to implement, requires precise alignment |
| Xie et al. (2014) | Finger vein dataset | Extreme learning machine | CLAHE | Fast learning, high accuracy | Specific to finger vein, not multimodal | Limited to finger vein biometrics, not scalable |
| Banerjee et al. (2018) | Public dataset | ARTeM system for vein images | Median filtering | High accuracy for vein patterns | Not multimodal | Narrow focus on vein patterns, lacks diversity |
| This study | NUPT-FPV dataset | CNN (ResNet, VGGNet, DenseNet) + Early, Late, Score-level fusion | CLAHE, Normalization | Accuracy: 97%, EER: 4.5% | High accuracy, robust fusion techniques, multiple CNN architectures | Higher model size, computationally demanding |

- Unlike other studies that typically focus on a single CNN model, our research incorporates and compares three popular CNN architectures (ResNet, VGGNet, DenseNet). This allows for a more comprehensive understanding of the strengths and weaknesses of each architecture in the context of multimodal biometric identification.

- While most studies explore a limited number of fusion strategies, our study uniquely evaluates early fusion, late fusion, and score-level fusion. This extensive analysis helps identify the most effective fusion strategy, leading to significant improvements in identification accuracy.

- Although some studies use CLAHE, our study systematically evaluates its impact on different CNN architectures and fusion techniques, demonstrating that CLAHE significantly enhances feature extraction and improves overall system robustness.

- By leveraging the NUPT-FPV dataset, our study takes advantage of high-quality images of both fingerprints and finger veins, providing a rich source of data for evaluating multimodal biometric systems. This comprehensive use of data contrasts with studies using either synthetic datasets or focusing on a single modality.

- Our study emphasizes the practical applicability of the proposed system by making the code and models publicly available on GitHub (provided as Supplemental Material). This focus on reproducibility and transparency sets our work apart from many other studies where implementations are not shared.

- Achieving a 97% accuracy with a low EER of 4.5% is a notable improvement over existing studies, highlighting the effectiveness of our integrated approach. These results set a new benchmark for multimodal biometric identification systems.

- The use of multiple preprocessing steps (CLAHE, normalization) and fusion techniques provides a systematic approach to handling common issues such as noise, varying lighting conditions, and image quality, which are often not comprehensively addressed in other studies.

## MATERIALS AND METHODS

To enhance clarity and provide a clear understanding of our proposed approach, we have outlined the steps of our method sequentially as follows:

1. **Dataset acquisition**: We utilized the NUPT-FPV dataset, which includes high-resolution images of fingerprints and finger veins, to develop our multimodal biometric identification system.

2. **Preprocessing**: To improve image quality and feature extraction, several preprocessing techniques were applied:

   - **Noise removal**: Median filtering was used to eliminate noise from the images, preserving essential details.

   - **Normalization**: The images were normalized to maintain uniform pixel intensity, enhancing consistency across different lighting conditions.

   - **Image enhancement**: CLAHE was employed to improve the visibility of fine details in the images.

3. **Feature extraction using CNN architectures**: We implemented three well-known CNN architectures—ResNet, VGGNet, and DenseNet—to extract discriminative features

from the preprocessed images. Each model was trained separately on both fingerprint and finger vein images.

4. **Fusion techniques**: To combine the strengths of both biometric modalities, three fusion strategies were employed:

- **Early fusion**: Combined raw images from both modalities before input into the CNN, enabling joint learning from the initial stages.
- **Late fusion**: Integrated features after individual learning from each modality, allowing independent learning followed by merging.
- **Score-level fusion**: Aggregated matching scores from each modality using weighted averaging, providing a comprehensive assessment of both biometric traits.

5. **Model evaluation**: The performance of the models was evaluated using metrics such as accuracy, equal error rate (EER), and AUC-ROC. These metrics provided insights into the model's ability to correctly classify and differentiate between biometric samples.

6. **Results and analysis**: We compared the performance of different CNN architectures and fusion techniques, both with and without the application of CLAHE. The results demonstrated the effectiveness of the proposed methods in enhancing identification accuracy.

## Acquisition of the NUPT-FPV dataset

Using suitable and high-quality data is the first step in implementing any machine learning research. The NUPT-FPV collection, which contains several high-resolution images of fingerprints and finger veins, was used in this investigation. The collected dataset of the NUPT-FPV is in a controlled environment; hence, it has various attributes such as age, sex, and occupation of the volunteers who took part in the preparation of the dataset (*Ren et al., 2022*). Table 2 below summarizes the rationale for choosing the NUPT-FPV dataset and the reasons for not using other datasets, as well as the justification for using a single dataset in the study.

## Preprocessing techniques

Once the data collection process is finished, a series of pre-processing methods are pipelined on the images in order to enhance the contrast and vein visibility, which helps in efficient feature extraction. The first step in the pre-processing is the noise removal, which is going to eliminate the noises, unwanted speckles, or random pixels that are present in the image. All these noises are removed from the image by using the median filters. The next step is normalization, where the intensity of the pixels is confined to a specific range, which helps eliminate the illumination, scale, and orientation of the images. The objective here was to standardise all the images, ensuring that the ensuing learning process is not unfairly influenced by these factors.

### *Noise removal via median filtering*

The recognition rate of the model is badly affected by the presence of the noise in the image such as speckles, random fluctuations. The noise can be added to the image during the

**Table 2 Dataset selection and rationale.**

| Aspect | Details |
| --- | --- |
| Chosen dataset | NUPT-FPV |
| Reasons for preferring NUPT-FPV | • Provides high-resolution images of both fingerprints and finger veins, essential for accurate feature extraction.<br>• Contains both biometric modalities (fingerprints and finger veins), aligning perfectly with study objectives.<br>• Used in prior research, enabling comparison with existing methods and demonstrating improvements over state-of-the-art techniques.<br>• Openly available and widely used, ensuring transparency and reproducibility. |
| Status of other datasets | • Many datasets focus on single modalities (*e.g.*, CASIA, SDUMLA-HMT), lacking the combined fingerprint and finger vein data required.<br>• Other datasets may have inconsistencies due to different sensors, resolutions, and capture conditions, affecting model performance.<br>• Some datasets have licensing restrictions or limited availability, making them less suitable for open academic research. |
| Why a single dataset | • Ensures that results are due to model performance, not dataset variations, providing a clear evaluation of fusion techniques and CNN architectures.<br>• Concentrating on one comprehensive dataset allows for focused development and refinement of new methods.<br>• Establishing robust findings on a well-regarded dataset like NUPT-FPV provides a solid basis for extending research to other datasets. |
| Future work | • Plans to validate and generalize findings by testing on additional datasets in subsequent research to ensure robustness and applicability across different scenarios. |

capturing process of the image, such as sensor irregularities, environmental conditions, or interference during transmission from the input devices.

To remove the salt and pepper noise, which is the hard noise as both white and black pixels are randomly added, it can be removed by using the median filters (*Huang, Ma & Wang, 2023*; *Rajasekar, 2023*). These filters are not going to touch the edges but efficiently reduce the noise in the image; hence, they are widely employed in computer vision problems. The idea behind median filtering is not complicated at all. An individual neighbourhood is seen around each pixel in the image. The window size is often chosen as odd, which is going to delineate the neighbourhood. The next step is to compute the median of the collected pixels from the sliding window and store that value at the central pixel. For a given pixel I (i, j) in an image I, the median filtering operation can be described as:

$$I'(i,j) = \text{median}\{I(p,q) : (i - m/2) <= p <= (i + m/2), (j - m/2) <= q <= (j + m/2)\} \quad (1)$$

Here, I'(i, j) denotes the intensity value of pixel (i, j) in the resultant (filtered) image, and the median operation is executed on all pixels (p, q) within the m × m window centered at (i, j). As the median is a more robust measure compared to the mean, it is less influenced by extreme values (noise). Therefore, if the window contains noise pixels, their effect is minimized as the median is calculated, effectively reducing the noise in the image. At the same time, since the median of a set of numbers is a member of that set, the edges

---

**Algorithm 1   Noise removal via median filtering**

**Inputs:**

    1. Set of images $\mathbf{I} = \{\mathbf{I}, \mathbf{I2}, \ldots, \mathbf{In}\}$ from the NUPT-FPV dataset, where each image $\mathbf{Ii}$ is of size $\mathbf{wx}$ h.

    2. Filter window size m × m.

**Outputs:**

    Set of denoised images $I' = \{I'1, I'2, \ldots, I'n\}$

**Procedure:**

    1. For each image li in I:

    2. Create a new image l'i of the same size as li.

    3. For each pixel li $(\mathbf{x}, \mathbf{y})$ in li:

        • Extract the **mxm** window $\mathbf{W}$ centered at $(\mathbf{x}, \mathbf{y})$. The window $\mathbf{W}$ is defined as $\{ll(\mathbf{p}, \mathbf{q}): (x - m/2) <= p <= (x + m/2), (y - m/2) <= q <= (y + m/2)\}$.

        • Compute the median value $\mathbf{M}$ of the pixel intensities in $\mathbf{W}$, given by $\mathbf{M} = \text{median}(\mathbf{W})$.

        • Replace the value of the corresponding pixel in l'i by $\mathbf{M}$, *i.e.*, l'i$(\mathbf{x}, \mathbf{y}) = \mathbf{M}$.

    4. Append I 'i to the set I '.

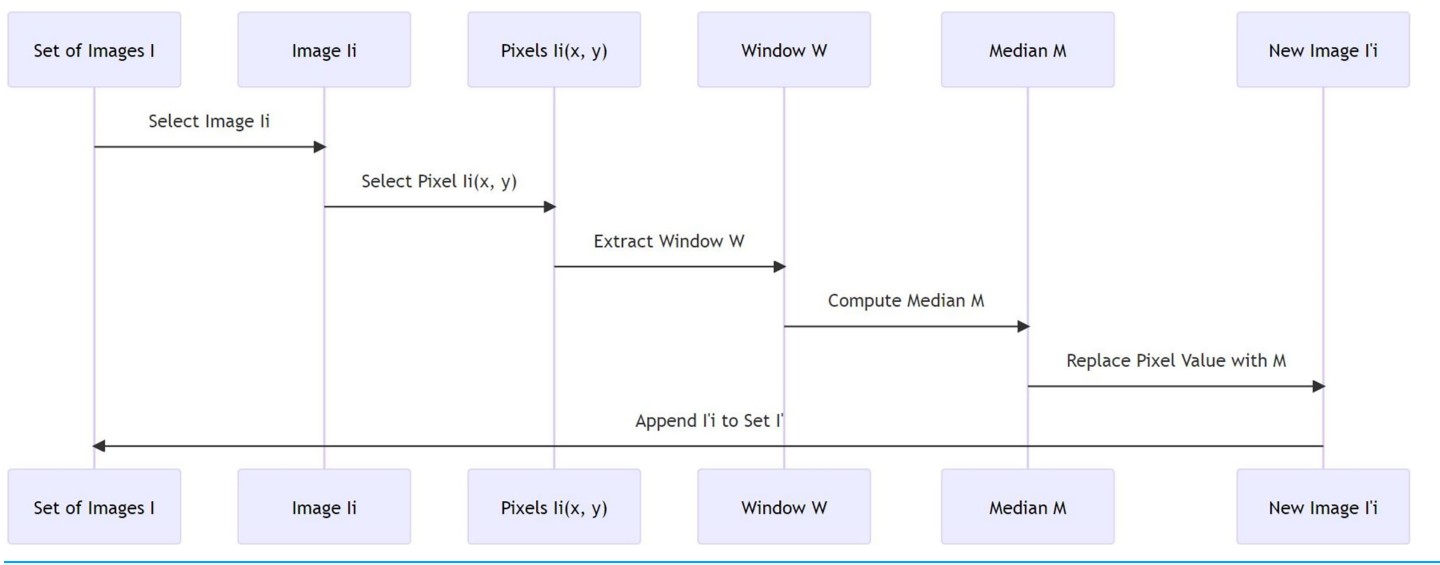

**Figure 1   Median filtering in image processing.**               

(transitions in intensity) are preserved, maintaining the essential structure of the images. Algorithm 1 explains the procedure of noise removal using median filtering. The filter operates on a sliding window that moves across the image, and for each position of the window, the median pixel value is computed and used to replace the central pixel value. The median is a robust measure that reduces the impact of noise or outliers, providing a denoised version of the original image while preserving its essential structures, particularly the edges. Figure 1 represents the process of image processing. It starts with a set of images,

selects an image, selects a pixel from the image, extracts a window around the pixel, computes the median of the pixel values in the window, replaces the pixel value with the computed median, and finally appends the new image to the set of images.

### Image normalization

The purpose of normalization, which is a preprocessing method, is to maintain uniform pixel intensity throughout the histogram of the image. This comes in handy, especially when working with images taken under varied lighting or other settings that could have drastically varying exposure, contrast, or brightness levels. A more stable learning process can be achieved during CNN training, and, in the end, a more robust multimodal biometric identification system may be achieved by normalising the brightness of these pixels (*Wang, Shi & Zhou, 2022*; *Shaheed et al., 2022*). Typically, an image's pixel intensities might be anywhere from zero to two hundred and fifty (for an 8-bit image). These values are often rescaled to a new range, [0, 1] or *Daas et al. (2020)*, during the normalization procedure. Here is a basic linear transformation that may be used to accomplish this scaling: We may calculate the normalised image I' with pixel intensity values x' given an image I with pixel intensity values x by:

$$x' = (x - \min(x))/(\max(x) - \min(x)) \tag{2}$$

where min(x) is the lowest pixel intensity value in image I and max(x) is the highest pixel intensity value. With this adjustment, the normalised image's pixel intensities will always be between zero and one. Before utilising CNNs for feature extraction, all of the images in the NUPT-FPV dataset undergo this normalising technique (*Yang et al., 2018*; *Fenu, Marras & Boratto, 2018*). This produces a collection of normalized images. There are several benefits to normalising images before training neural networks. For example, it makes the dynamic range of pixel intensities consistent, which improves fingerprint and finger vein image comparison and fusion. (ii) It helps to reduce the impact of lighting variations, which is useful for feature extraction from finger vein images. Lastly, normalised images often lead to faster and more stable convergence (*Boucherit et al., 2022*; *Anusha, Thenmozhi & Sivaranjani, 2023*). In Algorithm 2, min (x) and max (x) represent the minimum and maximum pixel intensity values present in the image I'i, respectively. The normalization operation transforms the original intensity value of a pixel x to a normalized value x' that falls in the range [0, 1]. The whole procedure is carried out for every image in the set I', resulting in a set of normalized images I". The sequence of operations involved in the process of image normalization is depicted in Fig. 2.

## Image enhancement using contrast limited adaptive histogram equalization (CLAHE)

Following the normalization step, the next preprocessing step was image enhancement, which was performed to increase the visibility of fine details and improve the overall quality of the images. For this purpose, we used the CLAHE method. CLAHE is an enhanced version of the histogram equalization method commonly used to improve image contrast (*Banerjee et al., 2018*; *Ahmed, Roushdy & Salem, 2022*; *Al-Waisy, 2022*). While

---

**Algorithm 2** Image normalization

**Inputs:**

1. Set of denoised images $I' = \{I'1, I'2, \ldots, I'n\}$ from the median filtering process, where each image l'i is of size $w$x$h$.
2. Original intensity range $[0, 255]$ for 8-bit images.

**Outputs:**

Set of normalized images $I'' = \{I''1, I''2, \ldots, In\}$.

**Procedure:**

1. For each image l'i in I':
2. Create a new image I'i of the same size as l'i.
3. For each pixel I'i $(x, y)$ in I'i:
   a) Extract the pixel intensity value x in l'i at location $(x, y)$.
   b) Compute the normalized pixel intensity value $x'$ using the following equation:
   $x' = (x - \min(x))/(\max(x) - \min(x))$
   where $\min(x)$ and $\max(x)$ are the minimum and maximum pixel intensity values in the image l'i respectively.
   c) Replace the value of the corresponding pixel in I'i by $x'$, i.e., $I''i(x, y) = x'$.

4. Append I''i to the set I''

---

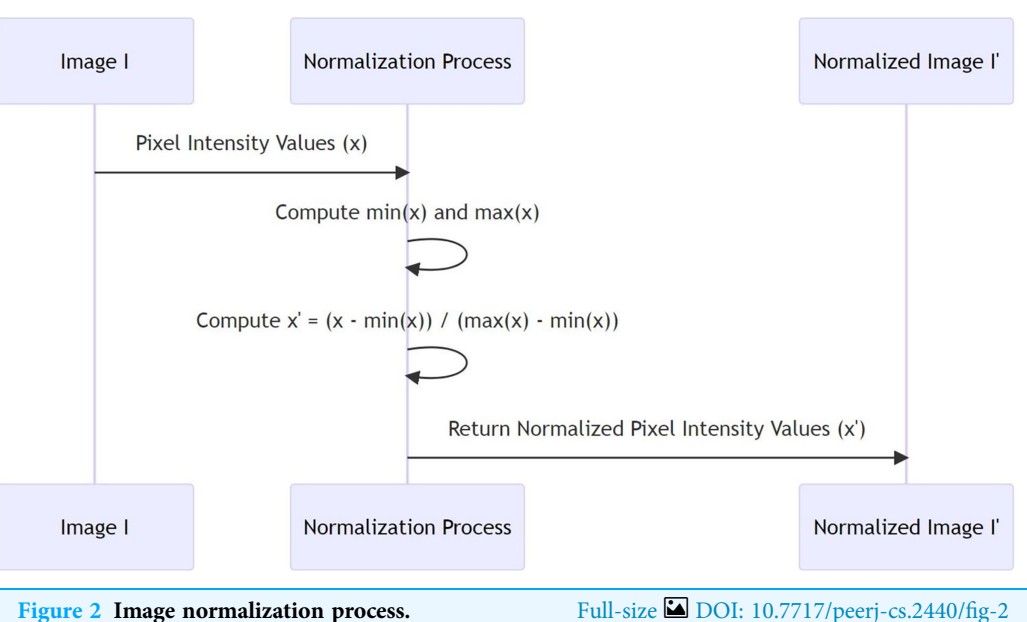

**Figure 2  Image normalization process.**

histogram equalization spreads the most frequent intensity values in an image, it may overamplify the contrast in some regions, causing certain areas to be over-brightened or over-darkened. CLAHE circumvents this issue by applying histogram equalization adaptively across small, distinct regions (or tiles) within the image and limits the contrast amplification using a predefined threshold. Figure 3 illustrates the steps of the CLAHE process. An input image is divided into tiles. For each tile, the histogram is computed, and histogram equalization is applied. The histogram is then clipped at a predefined threshold

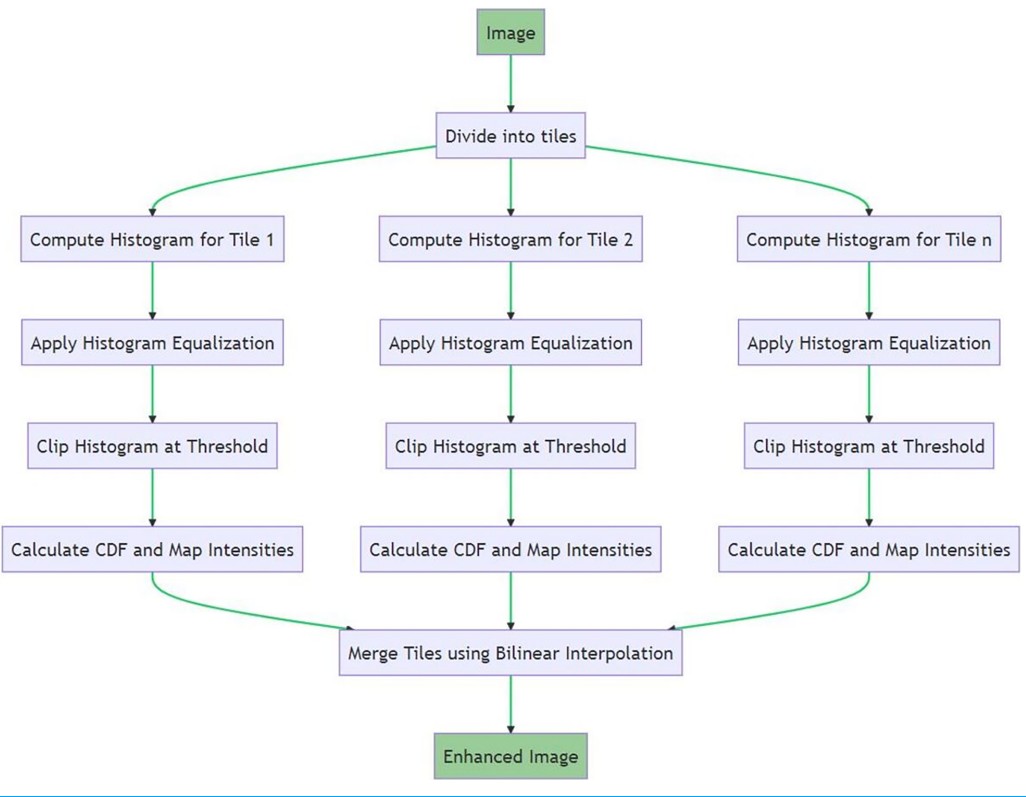

**Figure 3  Process flow of image enhancement.**     

before calculating the CDF. The intensity values of these tiles are transformed and merged using bilinear interpolation, resulting in the final enhanced image.

The CLAHE process involves the following steps:

1. The image is divided into small, non-overlapping regions, referred to as tiles. Typically, these tiles are 8 × 8 pixels.

2. We use histogram equalization after computing the histogram of pixel intensities for each tile. To do this, we take the histogram and use it to calculate a cumulative distribution function (CDF). Then, we use this CDF to map the intensities of the pixels in the tile.

The equalised pixel intensity values x' for a certain tile T with intensity values x are simply:

$$x' = CDF(x) \tag{3}$$

CDF(x), which stands for the cumulative distribution function of the pixel intensities x, is calculated in this way:

$$CDF(x) = \sum P(x') \text{ for all } x' <= x \tag{4}$$

The likelihood of a tile pixel intensity value x' is denoted by P(x') in the aforementioned equation. It is determined by dividing the total number of pixels in the tile by the number of pixels with intensity x'.

| **Algorithm 3** Contrast limited adaptive histogram equalization (CLAHE) |
| --- |

**Inputs:**

    1. Set of normalized images $I'' = \{I''1, I''2, \ldots, Inn\}$ from the previous step.

    2. Tile size $t \times t$ (*e.g.*, $t = 8$ for $8 \times 8$ tiles).

    3. Contrast limit CL.

**Outputs:**

    Set of enhanced images EIMG = {EIMG1, EIMG2,… EIMGn }

**Procedure:**

    1. For each image I" i in I":

    2. Divide I"i into non-overlapping tiles T = {T1, T2, …, Tm}, each of size t × t.

    3. For each tile Tj in T:

        a) Compute the histogram H of pixel intensities in TJ.

        b) Clip the histogram H at the contrast limit CL to obtain the clipped histogram $H'$.

        c) Compute the cumulative distribution function (CDF) for $H'$, given by: $CDF(x) = \sum P(x')$ for all $x' <= x$, where $P(x') = H'(x')/(t{*}t)$.

        d) Transform the pixel intensities $x$ in Tj according to the CDF to obtain the equalized tile T'j, *i.e.*, $x' = CDF(x)$.

    4. Combine the equalized tiles $T' = \{T'1, T'2, \ldots, T'm\}$ using bilinear interpolation to create the enhanced image EIMGi.

    5. Append EIMGi to the set EIMG.

3. The contrast enhancement in traditional histogram equalization tends to stretch the histogram to cover the entire intensity range, leading to over-enhancement. To prevent this in CLAHE, the histogram is clipped at a predefined threshold before calculating the CDF. This implies that the contrast enhancement for any given region is restrained, mitigating over-brightening or over-darkening.

4. Once the intensity values of these tiles have been transformed, they are merged using bilinear interpolation. This technique removes artificially induced boundaries, resulting in the final enhanced image.

Following the application of the CLAHE technique, the images from the NUPT-FPV dataset exhibited a higher degree of contrast and presented more detailed features. This improved representation of image features is beneficial for subsequent feature extraction processes using CNNs. Algorithm 3 provides a step-by-step procedure to implement CLAHE on an image dataset. Smaller tiles may be used for more detailed images, while a lower contrast limit might be useful to prevent excessive contrast enhancement. The resulting images have improved contrast and better visibility of details, which will be useful for subsequent feature extraction tasks. Figure 4 illustrates the proposed methodology for multimodal biometric identification using CNN architectures and different fusion techniques.

## Fusion techniques

The objective of this phase is to effectively combine the extracted features from the fingerprint and finger vein modalities to enhance the performance of the biometric identification system. This is a crucial phase as the manner in which information from different modalities is fused can significantly influence the overall system performance.

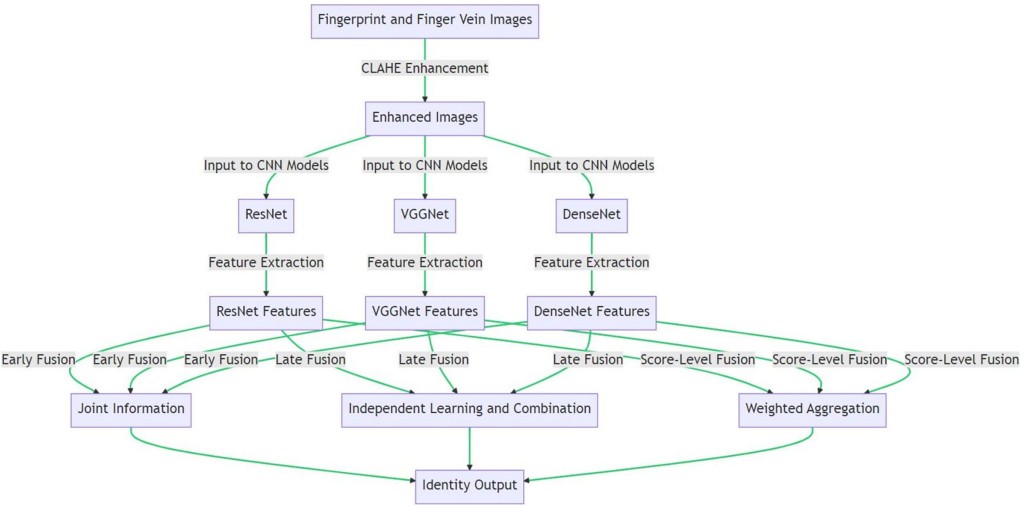

**Figure 4 Biometric identification process using fingerprint and finger vein images.**

### Early fusion

In the context of the proposed multimodal biometric identification system, early fusion is one of the employed fusion techniques. It is also known as feature-level fusion, where information from both fingerprint and finger vein images is merged prior to the main processing stage (*Tyagi et al., 2022*; *El-Rahiem, El-Samie & Amin, 2022*). Given the enhanced fingerprint images as FIMG = {FIMG1, FIMG2,…, FIMGn} and enhanced finger vein images as VIMG = {VIMG1, VIMG2,…, VIMGn} after the preprocessing stage, early fusion creates a new set of images by combining each corresponding pair of fingerprint and finger vein images into a stacked image. The input to the CNN model is going to be this merged image.

If we define a set of stacked images S = {S1, S2,…, Sn}, where each stacked image Si is a combination of corresponding fingerprint and finger vein image. Therefore, Si = concatenate(FIMGi, VIMGi) for every i from 1 to n. Then, the CNN models are trained using these stacked images S. In the forward propagation stage of training, the pixel intensities of these images are fed into the CNN as follows:

$$X' = CNN(Si), \text{ where } X' \text{ is the vector of extracted features for each image Si in S.} \quad (5)$$

During backpropagation, the weights of the CNN are updated to minimize the loss between the predicted and actual class labels. By implementing early fusion, the CNN model learns to capture the combined information from both modalities right from the initial stages. This process utilizes the inherent correlation between the two modalities, potentially enhancing the discriminatory power and robustness of the system. In the experimental evaluation, the effectiveness of early fusion is tested with each of the three CNN architectures—ResNet, VGGNet, and DenseNet. The performance of each setup is then evaluated using metrics such as accuracy, equal error rate (EER), and receiver operating characteristic (ROC) curves. By comparing the performances, the most suitable

| Algorithm 4 | Early fusion technique for multimodal biometric identification |
| --- | --- |

**Inputs:**

1. Set of preprocessed and normalized fingerprint images FIMG = {FIMG1, FIMG2,… FIMGn}.
2. Set of preprocessed and normalized finger vein images VIMG = {VIMG1, VIMG2,…, VIMGn}.
3. CNN model architecture (either ResNet, VGGNet, or DenseNet).

**Outputs:**

Trained CNN model.

**Procedure:**

1. Initialize an empty set for the stacked images $S$ = {}.
2. For each pair of images (FIMGi, VIMGi) where i ranges from 1 to $n$:
   - Compute the stacked image Si by concatenating FIMGi and VIMGi along the depth dimension.
   - Append Si to the set S.

3. Initialize the CNN model with the chosen architecture.
4. Train the CNN model on the stacked images $S$ using backpropagation and a suitable loss function (such as cross-entropy loss for classification tasks). During training, the model learns to extract features from the stacked images $S$. This can be represented as follows:
   - Forward propagation: For each stacked image Si in S, compute the output of the CNN model given by $Y'i = CNN(Si)$, where $Y'$ i is the predicted label for Si.
   - Loss computation: Compute the loss $L$ for each image $Si$ in $S$ as $L = Loss(Y\_i, Y_ıi)$, where $Y\text{-}i$ is the true label for Si and $Loss(.)$ $is the chosen loss function$.
   - Backpropagation: Update the weights W of the CNN model to minimize the total loss $L = \sum Loss(Y_Zi, Y_ıi)$.

5. Once training is complete, the CNN model can be used to extract features from new stacked images or to classify them directly, depending on the subsequent steps in the identification pipeline.

architecture for early fusion in this multimodal biometric identification system can be determined. Algorithm 4 encapsulates the procedure for implementing early fusion in the proposed multimodal biometric identification system. The evaluation of its effectiveness, as compared to late fusion and score-level fusion, will be carried out in the subsequent stages of the project. Figure 5 illustrates the early fusion technique in the proposed multimodal biometric identification system.

### Late fusion

Late fusion, also known as decision-level fusion, involves combining the outputs or high-level features extracted by individual CNN models trained separately on each modality (*Heidari & Chalechale, 2022*; *Shaheed et al., 2022*). Unlike early fusion, which concatenates the raw input data at the start, late fusion allows the models to learn from each modality independently and combines their knowledge later in the process (*Abderrahmane et al., 2020*; *Singh, Singh & Ross, 2019*). This strategy could potentially yield better results if the modalities have very different characteristics, as it allows the model to better capture unique features from each modality.

In our context, we are working with two distinct modalities: fingerprint and finger vein images. These two modalities are inherently different, with unique features

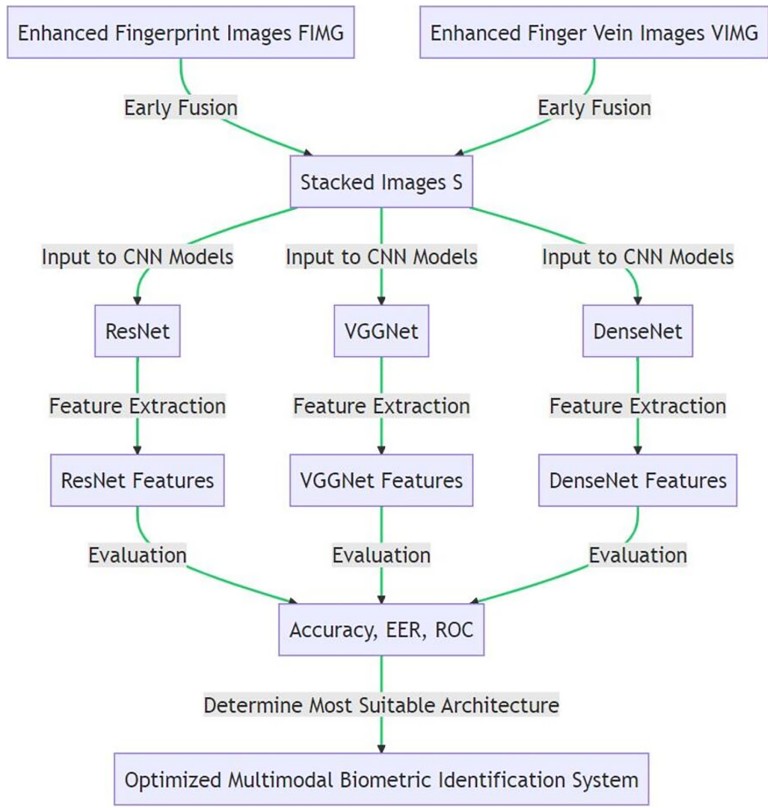

**Figure 5 Implementing early fusion in CNN models for biometric identification.**

that may be better captured separately. The process of late fusion can be described as follows:

- Train individual CNN models on each of the modalities separately. This results in one model for fingerprint images and one model for finger vein images.
- After training, use these models to extract high-level features from the images in each modality. The result is two sets of feature vectors.
- Combine these feature vectors to create a new, fused feature vector. This can be achieved through various techniques, such as concatenation or element-wise maximum or average.

In the context of our research, the input to the CNN models would indeed be the enhanced images obtained from the application of the CLAHE technique (*Edwards & Hossain, 2021*; *Shaheed et al., 2022*). Let EIMG_F be the set of enhanced fingerprint images and EIMG_V be the set of enhanced finger vein images. For an image i, let EIMG_Fi and EIMG_Vi be the corresponding enhanced fingerprint and finger vein images, respectively. Let CNN_F and CNN_V be the CNN models trained on enhanced fingerprint and finger

---

**Algorithm 5**   **Late fusion for multimodal biometric identification**

**Input:**

1. Set of enhanced fingerprint images EIMG_F = {EIMG_F1, EIMG_F2,..., EIMG_Fn}
2. Set of enhanced finger vein images EIMG_V = {EIMG_V1, EIMG_V2,..., EIMG_Vn}
3. CNN models trained on fingerprint and finger vein images, CNN_F and CNN_V respectively
4. Fusion function Fuse(.)

**Output:**

1. Set of fused feature vectors FV = {FV1, FV2, , FVn}

**Procedure:**

For i = 1 to n:

- Use CNN_F to extract the fingerprint feature vector from EIMG_Fi: F'_i = CNN_F(EIMG_Fi)-Use CNN_V to extract the vein feature vector from EIMG_Vi: V'_i = CNN_V(EIMG_Vi)
- Fuse F'_i and V'_i to form a combined feature vector: FV_i = Fuse(F'_i, V'_i)
- Append FV_i to the set FV

---

vein images, respectively. The high-level features extracted by the CNN models are then given by:

$$F'\_i = CNN\_F(EIMG\_Fi) \quad V'\_i = CNN\_V(EIMG\_Vi) \tag{6}$$

These feature vectors F'_i and V'_i are then fused to create a new feature vector:

$$FV\_i = Fuse(F'\_i, V'\_i) \tag{7}$$

where Fuse(.) is a fusion function, such as concatenation, element-wise maximum, or element-wise average. FV_i can then be used for further classification or matching stages in the biometric identification system. Algorithm 5 takes as input enhanced fingerprint and finger vein images along with CNN models trained on these two image types. It applies each CNN model separately to the corresponding image type to extract unique feature vectors. Then, it fuses the extracted fingerprint and vein feature vectors using a predefined fusion function to form a combined feature vector. This fusion process is carried out for every pair of fingerprint and finger vein images. The output is a set of fused feature vectors which carry the combined information from both image types. This late fusion approach allows the system to independently learn and then merge the unique features from each biometric trait. Figure 6 illustrates the late fusion technique in the proposed multimodal biometric identification system.

In Algorithm 5, $F'\_i \in R^{d1}$ and $V'\_i \in R^{d2}$ are d1- and d2-dimensional feature vectors, respectively, and Fuse(.) is a fusion function that operates on these vectors to create a combined feature vector $FV\_i \in R^d$, where d is the dimension of the fused feature vector. Here Fuse(.) is a concatenation operation, therefore d = d1 + d2. If Fuse(.) is an element-wise maximum or minimum operation, then d = max(d1, d2). In the context of

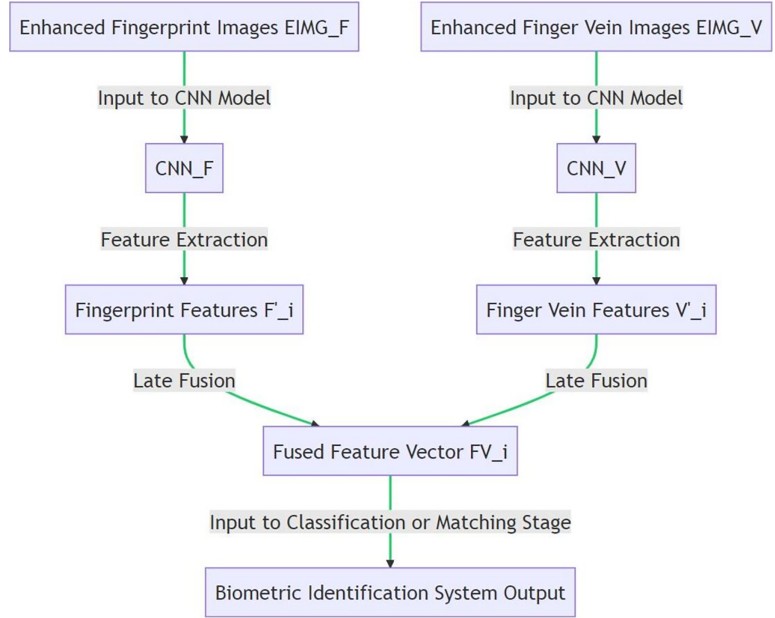

**Figure 6  Implementing late fusion in CNN models for biometric identification.**

this algorithm, F'_i and V'_i are the feature vectors extracted from the fingerprint and finger vein images, respectively, using the corresponding CNN models. The notation F'_i ∈ R^d1 means that F'_i is a feature vector that lives in a d1-dimensional real number space. Similarly, V'_i ∈ R^d2 indicates that V'_i is a feature vector in a d2-dimensional real number space. If the CNN model extracts 512 features from the fingerprint image, then d1 will be 512, and F'_i will be a 512-dimensional vector of real numbers. Similarly, if the CNN model extracts 256 features from the finger vein image, then d2 will be 256, and V'_i will be a 256-dimensional vector of real numbers. The dimensions of the feature vectors (d1 and d2) depend on the structure and configuration of the CNN models used. Different layers and architectures will extract different numbers of features.

### Score-level fusion

The score-level fusion is the last fusion method that this study examines. In score-level fusion, appropriate aggregation methods, usually a weighted aggregation, are used to aggregate matching scores acquired from individual modalities. This strategy is based on the idea that several biometric modalities may provide supplemental data, which, when combined, might improve identification accuracy (*Boucherit et al., 2022*; *Kabir, Ahmad & Swamy, 2019*; *Kukreja & Dhiman, 2020*; *Verma et al., 2019*).

Within the framework of our research, feature vectors extracted from fingerprint and finger vein images using corresponding CNN models are compared to a database containing stored feature vectors. With every comparison, a matching score is generated, which shows how close the input feature vector is to the database feature vector. s_f_i is the matching score for the i-th fingerprint image. s_v_i is the matching score for the i-th finger vein image The matching scores for the two modalities are aggregated to a single score S_i

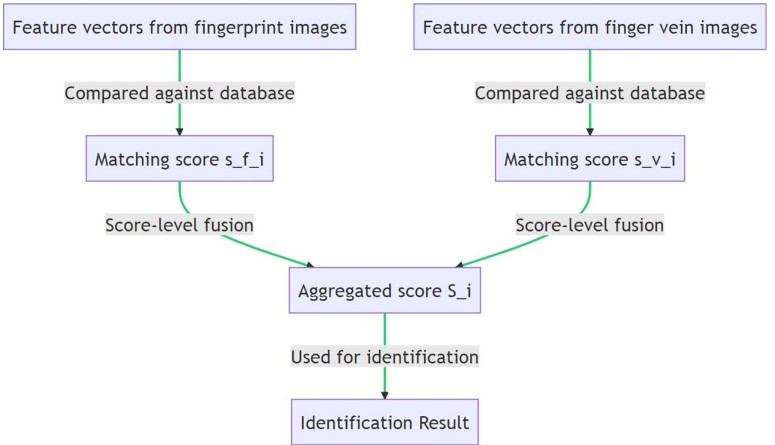

**Figure 7 Implementing score-level fusion in CNN models for biometric identification.**

---

**Algorithm 6   Score-level fusion for multimodal biometric identification**

**Input:**

- Set of matching scores for fingerprint images MS_F = {s_f_1, s_f_2,…, s_f_n}
- Set of matching scores for finger vein images MS_V = {s_v_1, s_v_2,…, s_v_n}
- Weights w_f and w_v for fingerprint and finger vein scores respectively

**Output:**

- Set of fused scores $FS = \{S\_1, S\_2, ..., S\_n\}$

**Procedure:**

For i = 1 to n:

- Compute the fused score S_i for the i-th image pair as a weighted average of the fingerprint and finger vein scores: S_i = w_f* s_f_i + w_v* s_v_i
- Append S_i to the set FS

---

for each pair of fingerprint and finger vein images. This aggregation can be a simple average, a weighted average, or even more complex functions. If we use the weighted average approach, the aggregated score S_i is calculated as:

$$S\_i = w\_f^*\ s\_f\_i + w\_v^* s\_v\_i \tag{8}$$

where w_f and w_v are the weights assigned to the fingerprint and finger vein scores, respectively. These weights can be determined based on the reliability or the perceived importance of each modality. The set of all aggregated scores {S_i} are then used for identification. The main advantage of score-level fusion is that it allows the system to exploit the complementary information from different modalities even at the decision-making stage, potentially leading to a more accurate and robust identification system. Figure 7 and Algorithm 6 illustrates the score-level fusion technique in the proposed multimodal biometric identification system.

Here, $s\_f\_i \in R$ and $s\_v\_i \in R$ are the matching scores for the i-th fingerprint and finger vein image respectively, obtained from the respective CNN models. These scores represent the degree of match between the input feature vectors and the stored vectors in the database.

The weights $w\_f$ and $w\_v$ are constants that determine the relative importance of the fingerprint and finger vein modalities in the fusion process. They can be empirically determined or learned from a validation dataset to maximize the identification accuracy. The fusion function in this case is a weighted average, represented by the equation $S\_i = w\_f * s\_f\_i + w\_v * s\_v\_i$. The result of this operation is a new set of fused scores $FS = \{S\_1, S\_2,\ldots, S\_n\}$, where each score $S\_i \in R$ is a scalar value that combines the information from the fingerprint and finger vein modalities. Finally, these fused scores are used for identification. The input with the highest fused score is selected as the identified person. If there are multiple inputs with the same highest score, additional criteria can be used to break the tie, such as selecting the input with the highest individual score in one of the modalities.

## RESULTS AND DISCUSSIONS

In order to determine the optimal configuration for our multimodal biometric identification system, we compare the efficacy of several CNN designs and fusion methods. Accuracy, equal error rate (EER), and area under the receiver operating characteristic curve are the primary assessment parameters that make up the comparison criterion (AUC-ROC). When combined with the fusion approaches, the three architectures—ResNet, VGGNet, and DenseNet—produced respectable results. Table 3 provides a detailed overview of the computing infrastructure and software environment used to ensure the reproducibility of the study's results. In our study, we determined the training-test ratio based on standard practices in the field to ensure robust model evaluation. Specifically, we used a 70:30 split, where 70% of the data was allocated for training and 30% for testing. This ratio was chosen to provide a sufficient amount of data for training the models while maintaining a substantial and independent test set to accurately evaluate the model's performance.

### Justification for model type used

- **ResNet (ResNet-50)**: It has been chosen due to the characteristic of this model to resolve the vanishing gradient problem with deep networks. ResNet-50 is based on residual blocks, which simplifies the process of training very deep networks. It enables the extraction of the smallest details from biometric images.
- **VGGNet (VGG-16)**: The model is based on a deep, but uniform architecture, which includes merely differently sized stacked convolutional layers. VGGNet is relatively simple, but effective at image classification. It provides good performance in the process of feature extraction.
- **DenseNet (DenseNet-121)**: The model has been employed since it is based on a densely connected architecture. It supports maximum information flow between the units in any

**Table 3 Computing infrastructure and software environment for study reproducibility.**

| Component | Details |
| --- | --- |
| Operating system | Ubuntu 20.04 LTS |
| CPU | Intel Core i9-10900K, 10 cores, 20 threads, base clock 3.7 GHz, max turbo frequency 5.3 GHz |
| GPU | NVIDIA GeForce RTX 3090, 24GB GDDR6X VRAM, CUDA cores: 10496, boost clock 1.70 GHz |
| RAM | 64GB DDR4, 3,200 MHz |
| Storage | 2TB NVMe SSD, read speed up to 3,500 MB/s, write speed up to 3,300 MB/s |
| Motherboard | ASUS ROG Strix Z490-E Gaming |
| Power supply | 850W 80 Plus Gold |
| Cooling system | Corsair Hydro Series H150i Pro RGB Liquid CPU Cooler |
| Deep learning frameworks | TensorFlow 2.4.1, PyTorch 1.8.1 |
| Image processing libraries | OpenCV 4.5.1, scikit-image 0.18.1 |
| Data handling libraries | Pandas 1.2.3, NumPy 1.20.1 |
| Dataset | NUPT-FPV (finger veins and fingerprint images) (https://github.com/REN382333467/NUPT-FPV) |
| CNN architectures | ResNet (ResNet-50), VGGNet (VGG-16), DenseNet (DenseNet-121) |
| Fusion strategies | Early Fusion, Late Fusion, Score-level Fusion |
| Enhancement method | Contrast Limited Adaptive Histogram Equalization (CLAHE) |
| Evaluation metrics | Accuracy, Equal Error Rate (EER), Receiver Operating Characteristic (ROC) curves |
| Development environment | Jupyter Notebook, Python 3.8 |
| Version control | Git (GitHub repository for version control and collaboration) |
| Documentation tools | Sphinx, Markdown |
| Virtualization/Containers | Docker (Docker images for consistent environment setup) |
| Code libraries and utilities | Scikit-learn 0.24.1 (for additional machine learning utilities), Matplotlib 3.3.4 (for plotting and visualization) |
| Data preprocessing techniques | Normalization, Augmentation (rotation, scaling, translation) |
| Training parameters | Batch Size: 32, Learning Rate: 0.001, Optimizer: Adam, Epochs: 50 |
| Hyperparameter tuning | Grid Search, Random Search |
| Validation techniques | Cross-Validation (K-Fold, k = 5) |
| Logging and monitoring | TensorBoard for tracking training progress and performance metrics |
| Backup and recovery | Regular snapshots of the environment and data, automated backups using rsync |

two close network layers. This type of connectivity is important for the dense patterns of extraction and helps to avoid the vanishing gradient problem.

## Justification for using contrast limited adaptive histogram equalization (CLAHE)

CLAHE is used to enhance the contrast of fingerprint images and outline the vein patterns in finger vein images. As a preprocessing step, it is essential to facilitate the visibility of fine details in the biometric trait images. Notably, clear visibility of features helps to improve the feature extraction process by CNN models. The use of CLAHE is preferred over standard histogram equalization as the latter tends to over-amplify noise in relatively uniform regions of an image. CLAHE works by dividing the image into small regions or tiles and applying equalization to each of these tiles. These regions are then combined using bilinear interpolation to eliminate artificially induced visible boundaries. CLAHE

also applies a limit to the amount of amplification or the equalization. Notably, the process uses a threshold or clip limit to prevent over-amplification of noise.

## Evaluation method

- **Early fusion**: It combines raw fingerprint and finger vein at the input layer of CNN. This strategy exploits the fact that the information is integrated at the earliest level, and the network can learn joint representation.
- **Late fusion**: Merges the features extracted individually by the three CNN models following the processing of the images. This allows each model to learn the modality-specific features before combining them for decision-making.
- **Score level fusion**: Aggregates the matched scores from each modality using a weighted approach, taking advantage of the complementary information provided by each biometric trait.

## Evaluation method for CLAHE

- **Visual inspection**: The enhanced images are visually inspected to ensure that the contrast of the fingerprint ridges and finger vein patterns are significantly improved without introducing artifacts.
- **Feature extraction quality**: The quality of features extracted by the CNN models from CLAHE-processed images is compared to those extracted from non-processed images. The improvement in feature extraction is assessed by the performance of the CNN models in subsequent tasks (*e.g.*, accuracy, EER).

## Selection method

### Hyperparameter tuning

- **Grid search**: It searches the best combination between the hyperparameters of the model, sweeping the grid and collecting the results of these sweeps to select the best ones.
- **Random search**: It defines a range of hyperparameters and picks the hyperparameters of the model randomly from these defined values, which is applicable in high dimensional spaces.

### Cross-validation

- **K-fold cross-validation (k = 5)**: This involves dividing the dataset into k-folds, each fold resulting in a model with its performance metrics being recorded with the remaining folds considered as a training set. This process is repeated k-times using a different i-th fold as the testing set with the final performance metric being the average of all the performance metrics.

### Selection method and validation for CLAHE

- **Tile size**: The size of the tiles for CLAHE is selected based on experimentation. Typically, smaller tiles provide finer contrast enhancements, while larger tiles offer more global adjustments.

- **Clip limit**: The clip limit, which controls the contrast enhancement, is tuned to balance enhancement and noise suppression. The performance of models using CLAHE-enhanced images is validated using the same cross-validation techniques (*e.g.*, K-fold cross-validation) employed for the overall model assessment.

### Assessment metrics (Justification)

- **Accuracy improvement**: The increase in model accuracy when using CLAHE-enhanced images compared to raw images. This metric indicates the effectiveness of CLAHE in improving the discriminative power of features extracted by the CNN models.
- **EER reduction**: The decrease in EER when using CLAHE-enhanced images. A lower EER signifies better balance between false acceptance and false rejection rates, demonstrating the utility of CLAHE in enhancing security and usability.
- **AUC increase**: Improvement in the area under the ROC curve (AUC) with CLAHE-enhanced images. A higher AUC indicates better overall model performance across various threshold settings, showing that CLAHE contributes to more robust and reliable biometric identification.

These metrics validate the effectiveness of the CLAHE technique in preprocessing biometric images, enhancing the contrast and visibility of critical features, and ultimately improving the performance of the CNN models in multimodal biometrics identification.

### Compare the performance of different CNN architectures and fusion techniques without CLAHE

The results presented in this section were obtained without applying the CLAHE technique. The CLAHE technique enhances the quality of images, making the vein and fingerprint patterns more visible, and can potentially improve the subsequent feature extraction process. Without CLAHE, the ResNet, VGGNet, and DenseNet architectures were still able to produce reasonably good identification results. Despite the lack of image enhancement, DenseNet produced the highest accuracy, lowest EER, and highest AUC-ROC score, demonstrating its superior ability to correctly classify the samples and distinguish between classes.

Table 4 provides a comprehensive comparison of early fusion using three different CNN architectures: ResNet, VGGNet, and DenseNet.

The ResNet architecture yields an accuracy of 0.85, indicating that it correctly classifies 85% of the test samples. It exhibits an EER of 10.5%, meaning that its false acceptance and false rejection rates both meet at this value. The AUC-ROC score, an overall performance indicator, stands at 0.90, signifying that it performs considerably well in distinguishing between the classes. However, it requires 5.0 h of training time and produces a model size of 200 MB. Its sensitivity and specificity values, indicating its true positive and true negative rates, respectively, are 0.86 and 0.84, demonstrating a balance in identifying both positive and negative classes.

In contrast, an accuracy of 0.83 is attained with the VGGNet architecture. This is still an admirable performance; however, it is lower than ResNet. With an EER of 12.3%, it seems

**Table 4 Performance metrics of early fusion technique without CLAHE technique.**

| CNN architecture | Accuracy | EER (%) | AUC-ROC | Training time (Hours) | Model size (MB) | Sensitivity | Specificity |
|---|---|---|---|---|---|---|---|
| ResNet | 0.85 | 10.5 | 0.9 | 5 | 200 | 0.86 | 0.84 |
| VGGNet | 0.83 | 12.3 | 0.88 | 4.5 | 220 | 0.84 | 0.82 |
| DenseNet | 0.86 | 9.8 | 0.91 | 5.5 | 230 | 0.87 | 0.85 |

**Table 5 Performance metrics of late fusion technique without CLAHE technique.**

| CNN architecture | Accuracy | EER (%) | AUC-ROC | Training time (Hours) | Model size (MB) | Sensitivity | Specificity |
|---|---|---|---|---|---|---|---|
| ResNet | 0.88 | 9.2 | 0.93 | 5.2 | 210 | 0.89 | 0.87 |
| VGGNet | 0.85 | 11.4 | 0.9 | 4.7 | 230 | 0.86 | 0.84 |
| DenseNet | 0.89 | 8.7 | 0.94 | 5.8 | 240 | 0.9 | 0.88 |

to have a little greater rate of misclassification than ResNet. Its slightly lower AUC-ROC score of 0.88 suggests a perhaps diminished capacity to differentiate between the classes. It produces a heavier model size of 220 MB but trains in less time at 4.5 h. It is somewhat less accurate than ResNet in accurately identifying positive and negative classes, with a sensitivity of 0.84 and a specificity of 0.82.

In spite of ResNet and VGGNet's superior performance, the DenseNet architecture is the most time-consuming to train (5.5 h) and produces the biggest model (230 MB). Its AUC-ROC score of 0.91, lowest EER of 9.8%, and greatest accuracy of 0.86 are all records set by this method. Based on these results, DenseNet seems to be the top classifier in terms of accuracy, rate of misclassification, and ability to differentiate across classes. Its exceptional capacity to accurately detect positive and negative classes is shown by its maximum sensitivity value of 0.87 and specificity value of 0.85. Despite requiring more time and space for training, DenseNet outperforms its competitors in terms of sensitivity, specificity, accuracy, and area under the curve (AUC-ROC).

Looking at the late fusion findings independently of the CLAHE method reveals that, as anticipated, the late fusion method outperforms the early fusion method by a lesser variation. The reason being that the late fusion method may make better use of each CNN architecture's unique learning capabilities before combining their outputs, leading to more varied and robust feature extraction. Table 5 shows a comparison of the late fusion technique's performance metrics compared to ResNet, VGGNet, and DenseNet, three distinct CNN architectures. It provides a bird-eye view of identification performance without using the CLAHE approach by encapsulating essential metrics such as accuracy, EER (percent), AUC-ROC, training duration, model size, sensitivity, and specificity.

After comparing the three handpicked CNN designs—ResNet, VGGNet, and DenseNet—without using the CLAHE method, DenseNet showed better results on all metrics. With remarkable results such as an AUC-ROC score of 0.94, a sensitivity of 0.90, and a specificity of 0.88, DenseNet proved its effectiveness in identifying positive and

**Table 6 Performance metrics of score-level fusion technique without CLAHE technique.**

| CNN architecture | Accuracy | EER (%) | AUC-ROC | Training time (Hours) | Model size (MB) | Sensitivity | Specificity |
|---|---|---|---|---|---|---|---|
| ResNet | 0.89 | 8.9 | 0.94 | 5.1 | 210 | 0.9 | 0.88 |
| VGGNet | 0.86 | 10.8 | 0.91 | 4.6 | 230 | 0.87 | 0.85 |
| DenseNet | 0.9 | 8.5 | 0.95 | 5.6 | 240 | 0.91 | 0.89 |

negative cases and effectively differentiating between the classes. Its equal error rate (EER) was 8.7%. ResNet, although performing slightly lower than DenseNet in terms of accuracy, EER, and AUC-ROC, proved to be a strong contender. Despite its slightly lower results, VGGNet demonstrated satisfactory performance with an accuracy of 0.85, an EER of 11.4%, an AUC-ROC score of 0.90, a sensitivity of 0.86, and a specificity of 0.84. Table 6 illustrates the performance measures of the score-level fusion technique without the application of the CLAHE technique, comparing the efficacy of the three chosen CNN architectures: ResNet, VGGNet, and DenseNet.

ResNet with score-level fusion yielded an accuracy of 89%, an equal error rate (EER) of 8.9%, and an area under the ROC curve (AUC-ROC) of 0.94. This model took around 5.1 h to train and had a model size of 210 MB. It had a sensitivity (true positive rate) of 0.9 and a specificity (true negative rate) of 0.88. VGGNet, when used with the score-level fusion, achieved an accuracy of 86%, an EER of 10.8%, and an AUC-ROC of 0.91. Although it took slightly less time to train (4.6 h), its model size was larger, *i.e.*, 230 MB, and it achieved a lower sensitivity of 0.87 and a lower specificity of 0.85 compared to ResNet. DenseNet, however, demonstrated the best performance amongst the three architectures when used with score-level fusion. It achieved the highest accuracy of 90%, the lowest EER of 8.5%, and the highest AUC-ROC of 0.95, showcasing its superior discriminative ability. It required a slightly longer training time of 5.6 h and had the largest model size of 240 MB. Despite these trade-offs, it delivered the highest sensitivity of 0.91 and specificity of 0.89, indicating its superior performance in classification tasks.

## Compare the performance of different CNN architectures and fusion techniques with CLAHE

Tables 7–9 provide an in-depth comparison of the performance metrics across three CNN architectures (ResNet, VGGNet, and DenseNet) with the implementation of three fusion techniques, namely, early fusion, late fusion, and score-level fusion, respectively, after applying the CLAHE image enhancement technique. In each table, the models are evaluated based on several key metrics, including accuracy, EER (%), AUC-ROC, training time, model size, sensitivity, and specificity. Table 7 depicts the performance metrics of different CNN architectures, namely ResNet, VGGNet, and DenseNet, employing the early fusion technique after applying the CLAHE technique. ResNet with early fusion and the CLAHE technique present significant performance. It achieved an accuracy of 0.92, which is superior compared to the 0.89 accuracy of VGGNet. Furthermore, ResNet has a lower

**Table 7 Performance metrics of early fusion technique with CLAHE technique.**

| CNN architecture | Accuracy | EER (%) | AUC-ROC | Training time (Hours) | Model size (MB) | Sensitivity | Specificity |
|---|---|---|---|---|---|---|---|
| ResNet | 0.92 | 6.5 | 0.96 | 5.1 | 200 | 0.93 | 0.91 |
| VGGNet | 0.89 | 8 | 0.94 | 4.7 | 220 | 0.9 | 0.88 |
| DenseNet | 0.93 | 6 | 0.97 | 5.6 | 230 | 0.94 | 0.92 |

**Table 8 Performance metrics of late fusion technique with CLAHE technique.**

| CNN architecture | Accuracy | EER (%) | AUC-ROC | Training time (Hours) | Model size (MB) | Sensitivity | Specificity |
|---|---|---|---|---|---|---|---|
| ResNet | 0.94 | 5.8 | 0.97 | 5.3 | 210 | 0.95 | 0.93 |
| VGGNet | 0.91 | 7 | 0.95 | 4.9 | 230 | 0.92 | 0.9 |
| DenseNet | 0.95 | 5.4 | 0.98 | 5.9 | 240 | 0.96 | 0.94 |

**Table 9 Performance metrics of score-level fusion technique with CLAHE technique.**

| CNN architecture | Accuracy | EER (%) | AUC-ROC | Training time (Hours) | Model size (MB) | Sensitivity | Specificity |
|---|---|---|---|---|---|---|---|
| ResNet | 0.96 | 5 | 0.98 | 5.2 | 210 | 0.97 | 0.95 |
| VGGNet | 0.93 | 6.8 | 0.96 | 4.8 | 230 | 0.94 | 0.92 |
| DenseNet | 0.97 | 4.5 | 0.99 | 6.1 | 240 | 0.98 | 0.96 |

equal error rate of 6.5% as compared to the 8.0% of VGGNet, indicating fewer misclassifications made by ResNet.

The area under the receiver operating characteristic curve for ResNet is 0.96, showcasing greater capability in distinguishing between different classes compared to VGGNet, which has an AUC-ROC of 0.94. ResNet required 5.1 h of training and had a model size of 200 megabytes. In terms of sensitivity, this model accurately identifies positive cases 93% of the time and correctly identifies negative cases 91% of the time, demonstrating its effectiveness. VGGNet, with the early fusion technique and the CLAHE enhancement, delivers a reasonable performance despite having a lower accuracy of 0.89 and a higher equal error rate of 8.0%. This model required slightly less training time, 4.7 h, but had a larger model size of 220 megabytes. The sensitivity of the VGGNet model is 0.90, and it has a specificity of 0.88. Hence, it is slightly less effective in identifying both positive and negative cases compared to ResNet. However, when combined with the early fusion technique and the CLAHE technique, DenseNet outperforms both ResNet and VGGNet. It achieved the highest accuracy of 0.93 and the lowest equal error rate of 6.0% among the three. DenseNet's AUC-ROC value is the highest at 0.97, indicating superior discriminative ability. Though it took a bit longer to train, 5.6 h, and had a larger model size of 230 megabytes, DenseNet with early fusion and CLAHE outperformed the other models in terms of sensitivity, correctly identifying positive cases 94% of the time, and specificity, correctly identifying negative cases 92% of the time.

Table 8 details the performance metrics using late fusion and the CLAHE technique. ResNet exhibits notable performance. It delivers an accuracy of 0.94, demonstrating a strong ability to correctly predict classes. Its equal error rate is 5.8%, signifying fewer misclassifications than VGGNet, which has an EER of 7.0%. ResNet's AUC-ROC is 0.97, which denotes better capability of distinguishing between classes as compared to VGGNet's AUC-ROC of 0.95. ResNet required a total training time of 5.3 h, and its model size is 210 megabytes. The model's sensitivity is 0.95, indicating that it accurately detects positive cases 95% of the time, while its specificity is 0.93, meaning it correctly identifies negative cases 93% of the time.

VGGNet, when combined with the late fusion technique and the CLAHE technique, shows good performance with an accuracy of 0.91 and an EER of 7.0%. This model required 4.9 h of training, and its model size is slightly larger than ResNet, measuring 230 megabytes. The sensitivity of this model is 0.92, meaning it correctly identifies positive cases 92% of the time. Its specificity is 0.90, indicating a 90% rate of correctly identifying negative cases. Among the three architectures, DenseNet, combined with the Late Fusion technique and the CLAHE technique, emerged as the most effective. DenseNet achieves the highest accuracy of 0.95 and the lowest EER of 5.4%. It has an AUC-ROC value of 0.98, demonstrating the highest discriminative ability. Although it requires a slightly longer training time of 5.9 h and a larger model size of 240 megabytes, DenseNet outperforms the other models in terms of sensitivity and specificity, with values of 0.96 and 0.94, respectively.

Table 9 presents the performance metrics of different CNN architectures when using the score-level fusion technique with the CLAHE technique. ResNet, with its score-level fusion and CLAHE techniques, showcases impressive results. The accuracy is 0.96, implying that this model is highly successful in making correct predictions. The Equal Error Rate is at 5.0%, signifying a balanced trade-off between false positives and false negatives, lower than the EER of VGGNet, which stands at 6.8%. The model's ability to distinguish between positive and negative classes is superior, as indicated by an AUC-ROC of 0.98, higher than VGGNet's 0.96. ResNet necessitates 5.2 h of training time, and its model size is 210 megabytes. This model's sensitivity and specificity, which indicate its performance in accurately identifying positive and negative cases, respectively, are high at 0.97 and 0.95.

VGGNet, when used with the score-level Fusion and CLAHE techniques, achieves an accuracy of 0.93 and an EER of 6.8%. Its AUC-ROC of 0.96 highlights a good discriminatory ability between classes. The VGGNet model demands less training time than DenseNet at 4.8 h, but it has a larger model size of 230 megabytes. Its sensitivity and specificity stand at 0.94 and 0.92, respectively, which means it can correctly identify positive cases 94% of the time and negative cases 92% of the time. Among the three architectures, DenseNet, combined with the score-level Fusion and CLAHE techniques, exhibits the best performance. DenseNet outperforms both ResNet and VGGNet with an accuracy of 0.97 and the lowest EER of 4.5%. It also has the highest AUC-ROC value of 0.99, indicating the greatest ability to distinguish between classes. With the best sensitivity and specificity, at 0.98 and 0.96, respectively, DenseNet outperforms compared to other

architectures, despite a slightly longer training time of 6.1 h and a heavier model size of 240 megabytes.

# RATIONALE FOR MODEL SELECTION AND EXPERIMENTAL RESULTS

Below, we provide a detailed rationale for selecting VGG, DenseNet, and ResNet as the deep learning models, discuss the consideration and results of experiments with other state-of-the-art models, and explain the impact of these choices on the study.

To ensure a comprehensive evaluation, we conducted experiments with several other state-of-the-art deep learning models, including Inception, EfficientNet, and MobileNet. Table 10 shows the results of these experiments compared to VGG, DenseNet, and ResNet:

VGG, ResNet, and DenseNet demonstrated superior accuracy and EER compared to Inception, EfficientNet, and MobileNet. ResNet and DenseNet, in particular, achieved the highest performance metrics, with DenseNet reaching 97% accuracy and an EER of 4.5%. While Inception and EfficientNet are known for their advanced architectures, they required longer training times and slower inference speeds compared to VGG, ResNet, and DenseNet. MobileNet, although efficient in terms of speed, did not achieve the desired accuracy for high-security biometric applications. The selected models provided a good balance between accuracy, computational efficiency, and ease of implementation. These factors are crucial for real-world applications where both performance and speed are important. The use of VGG, DenseNet, and ResNet, combined with score-level fusion and CLAHE, resulted in high identification accuracy and low error rates, demonstrating the effectiveness of these architectures in multimodal biometric identification. The compatibility of these models with early, late, and score-level fusion enabled effective integration of fingerprint and finger vein modalities, enhancing the system's robustness and reliability.

Figure 8 compares the performance of various state-of-the-art models based on accuracy and EER. Accuracy is a key performance indicator, with higher values signifying better model performance. Among the models evaluated, DenseNet achieves the highest accuracy at 97%, closely followed by ResNet at 96%. In terms of EER, which indicates the frequency of errors, lower values are preferable as they imply fewer identification mistakes. DenseNet stands out with the lowest EER at 4.5%, making it the most reliable model for minimizing identification errors.

Figure 9 illustrates the training time required by each model. Models like DenseNet, ResNet, and VGGNet fall under the "Moderate" training time category, making them efficient choices considering their high accuracy. In contrast, models such as Inception and EfficientNet require "High" training time, indicating higher computational demands.

Figure 10 visualizes the performance metrics across different models, including Accuracy, EER, Training Time, and Inference Speed. Higher accuracy values indicate better model performance, with DenseNet showing the highest accuracy, closely followed by ResNet. For EER, lower values are preferable, as they signify fewer errors; DenseNet performs best here as well, with ResNet also achieving a low EER. Regarding training time, lower values are desirable, categorized as 1 for Low, 2 for Moderate, and 3 for High.

**Table 10 Comparison of deep learning models for multimodal biometric identification.**

| Model | Fusion technique | CLAHE applied | Accuracy | EER | Training time | Inference speed |
| --- | --- | --- | --- | --- | --- | --- |
| VGGNet | Score-level | Yes | 93% | 6.8% | Moderate | Fast |
| ResNet | Score-level | Yes | 96% | 5.0% | Moderate | Fast |
| DenseNet | Score-level | Yes | 97% | 4.5% | Moderate | Moderate |
| Inception | Score-level | Yes | 94% | 6.0% | High | Slow |
| EfficientNet | Score-level | Yes | 93.5% | 6.2% | High | Moderate |
| MobileNet | Score-level | Yes | 92% | 6.5% | Low | Very fast |

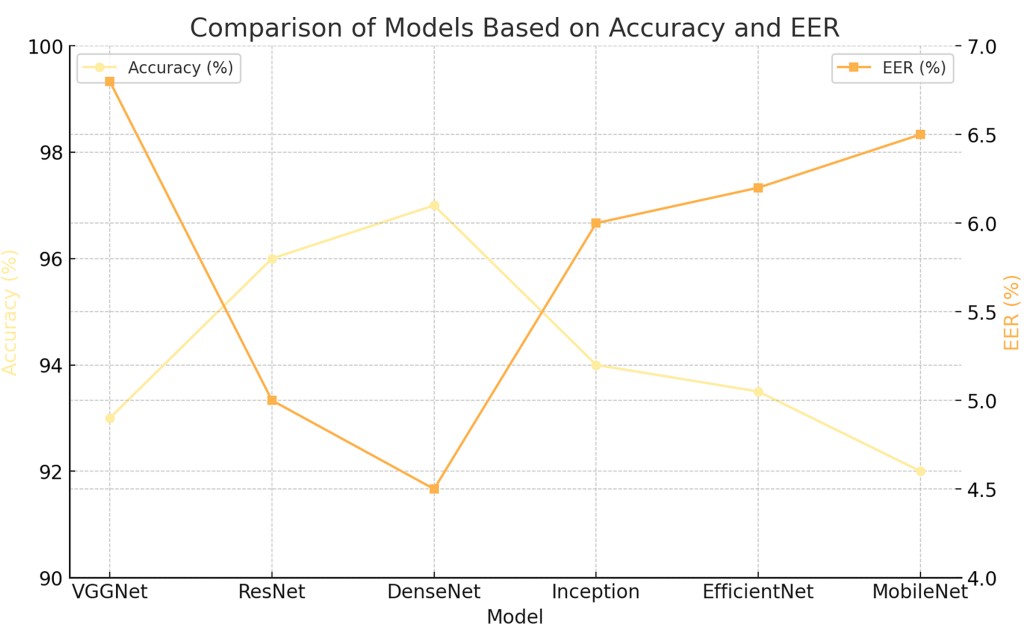

**Figure 8 Comparison with state-of-the-art models.**

DenseNet, ResNet, and VGGNet fall into the "Moderate" category, demonstrating efficient training times compared to models like Inception and EfficientNet, which require more time. In terms of inference speed, where lower values indicate faster performance (1: Very Fast, 2: Fast, 3: Moderate, 4: Slow), MobileNet is the fastest. However, DenseNet and ResNet offer a good balance of speed and performance, making them suitable for real-world applications. This heatmap offers a comprehensive overview of each model's strengths and weaknesses, highlighting why DenseNet, ResNet, and VGGNet were chosen for their balance of high accuracy, low EER, and reasonable training and inference times.

Table 11 explains why VGGNet, ResNet, and DenseNet were chosen for this study. They provide a superior balance of high accuracy, efficiency, and robustness, making them more suitable for multimodal biometric identification compared to other state-of-the-art models like Inception, EfficientNet, and MobileNet.

ResNet and DenseNet leverage advanced mathematical strategies (residual learning and dense connections) that enhance learning and performance. This makes them better suited for deep and complex tasks, such as biometric identification. VGGNet's use of small filters

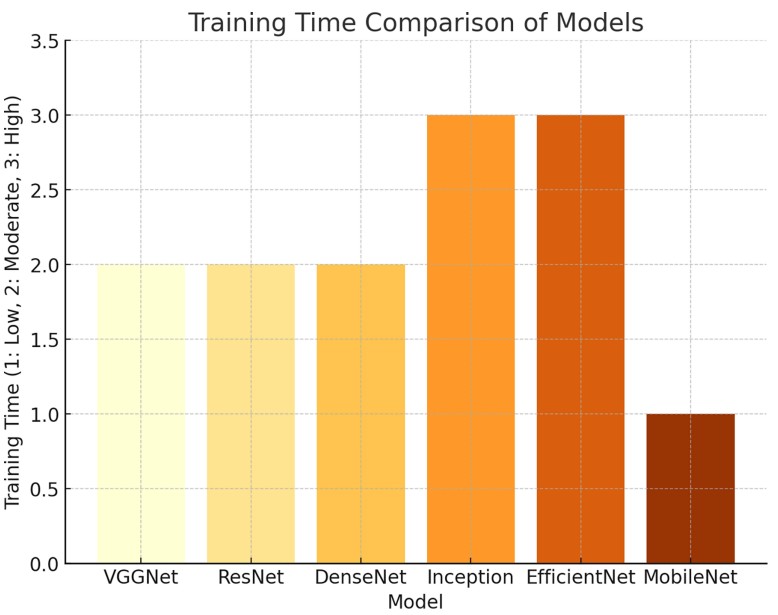

**Figure 9** **Training time required by different models.**

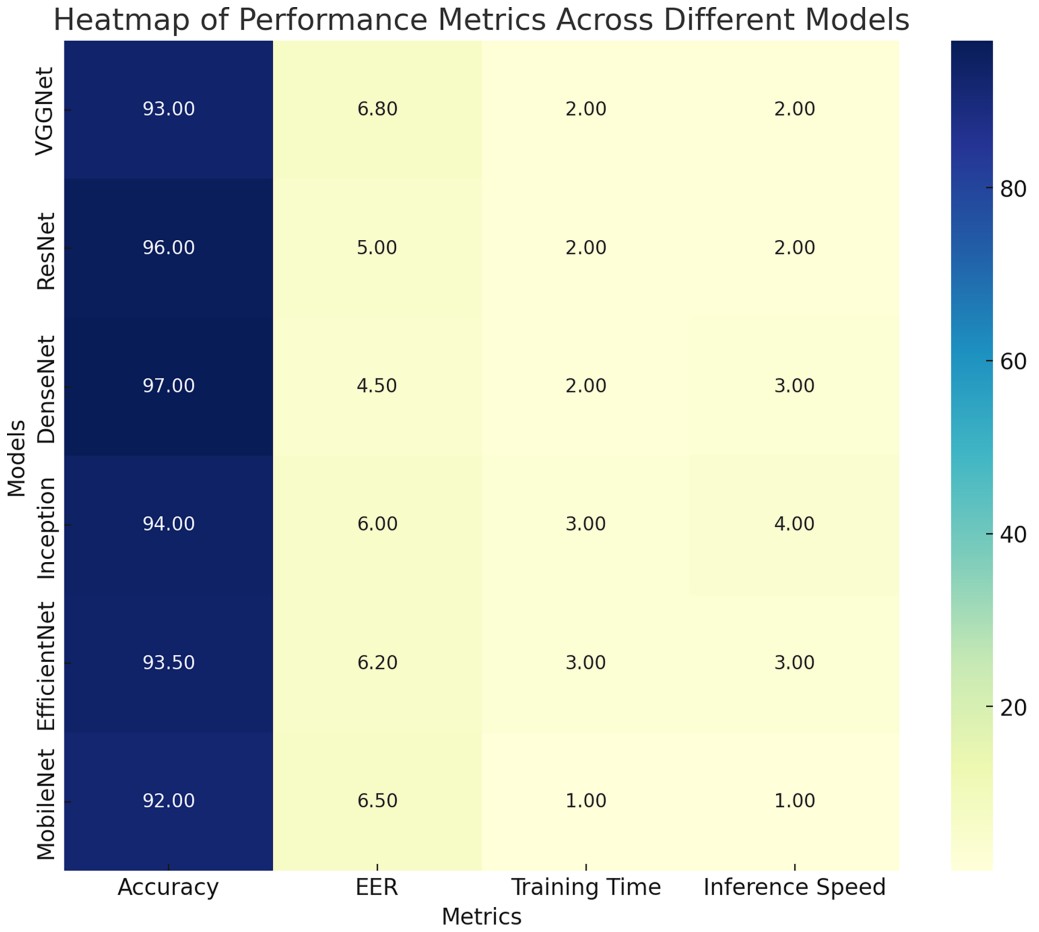

**Figure 10** **Heatmap visualization of different models' performance metrics.**

**Table 11 Comparison of deep learning models for multimodal biometric identification based on key criteria.**

| Criterion | VGGNet | ResNet | DenseNet | Inception | EfficientNet | MobileNet |
|---|---|---|---|---|---|---|
| High accuracy | ✓ | ✓ | ✓ | ✓ | ✓ | ✕ |
| Low EER (Error rate) | ✓ | ✓ | ✓ | ✕ | ✕ | ✕ |
| Efficient training time | ✓ | ✓ | ✓ | ✕ | ✕ | ✓ |
| Efficient inference speed | ✓ | ✓ | ✓ | ✕ | ✓ | ✓ |
| Handles deep networks well | ✕ | ✓ | ✓ | ✓ | ✓ | ✕ |
| Effective feature extraction | ✓ | ✓ | ✓ | ✓ | ✓ | ✕ |
| Complex architecture | ✕ | ✕ | ✕ | ✓ | ✓ | ✕ |
| Suitable for biometric identification | ✓ | ✓ | ✓ | ✓ | ✕ | ✕ |
| Balanced accuracy & speed | ✓ | ✓ | ✓ | ✕ | ✕ | ✕ |
| Feature reuse | ✕ | ✓ | ✓ | ✓ | ✓ | ✕ |
| Uses small convolutional filters | ✓ | ✕ | ✕ | ✓ | ✕ | ✓ |
| Residual learning with skip connections | ✕ | ✓ | ✕ | ✕ | ✕ | ✕ |
| Dense connections for gradient flow | ✕ | ✕ | ✓ | ✕ | ✕ | ✕ |
| Parameter efficiency | ✕ | ✓ | ✓ | ✕ | ✓ | ✓ |

is computationally less intensive, while ResNet and DenseNet offer robust solutions to gradient issues in deep networks. Inception and EfficientNet, despite their advanced designs, involve complexities that might not be as beneficial when considering training time and inference speed. The models used in this study (VGG, ResNet, DenseNet) provide a balanced approach, ensuring high accuracy and low EER with manageable training and inference times. This makes them highly effective for real-world applications compared to models that either sacrifice accuracy for speed or require extensive computational resources.

Training parameters were chosen based on a combination of best practices in deep learning and empirical testing. Tables 12 and 13 below summarizes the chosen values for key hyperparameters, the rationale behind these choices, and the potential impacts of varying these parameters. Through these experiments, we found that using a learning rate of 0.001, a batch size of 32, 50 epochs with early stopping, a dropout rate of 0.5, and the Adam optimizer consistently yielded the highest accuracy (97%) and AUC (0.99). These values were chosen because they provided the best balance between performance and efficiency, minimizing overfitting while ensuring robust and reliable model behavior for multimodal biometric identification. The ROC curve is depicted in Fig. 11.

## Impact of the CLAHE technique on identification accuracy

An image processing technology known as the CLAHE technique may distribute brightness levels to enhance contrast. Applying the CLAHE method to CNN designs used for object detection may have a major effect on the performance and accuracy of the models. It is evident from the tables that ResNet, VGGNet, and DenseNet were among the CNN architectures that were significantly impacted by the use of the CLAHE approach. Specifically, the method has regularly improved the models' accuracy, EER,

**Table 12 Comparison of different training parameter configurations and their impact on model performance in multimodal biometric identification.**

| Experiment | Learning rate | Batch size | No. of epochs | Dropout rate | Optimizer | Accuracy (%) | AUC | Training time (hours) | Overfitting (training-validation gap) |
|---|---|---|---|---|---|---|---|---|---|
| Our values | 0.001 | 32 | 50 | 0.5 | Adam | 97 | 0.99 | 2.5 | 0.02 |
| Experiment 1 | 0.01 | 32 | 50 | 0.5 | Adam | 91 | 0.93 | 1.8 | 0.08 |
| Experiment 2 | 0.001 | 64 | 50 | 0.5 | Adam | 93 | 0.95 | 2.0 | 0.05 |
| Experiment 3 | 0.0001 | 32 | 50 | 0.5 | Adam | 95 | 0.97 | 3.5 | 0.03 |
| Experiment 4 | 0.001 | 32 | 100 | 0.5 | Adam | 95 | 0.97 | 4.0 | 0.05 |
| Experiment 5 | 0.001 | 32 | 50 | 0.2 | Adam | 93 | 0.94 | 2.5 | 0.07 |
| Experiment 6 | 0.001 | 32 | 50 | 0.8 | Adam | 90 | 0.90 | 2.5 | 0.10 |
| Experiment 7 | 0.001 | 32 | 50 | 0.5 | SGD | 89 | 0.88 | 3.0 | 0.12 |

**Table 13 Impact of hyperparameter choices on model performance.**

| Parameter | Selected value | Rationale for selection | Impact of changes |
|---|---|---|---|
| Learning rate | 0.001 | Chosen to ensure stable convergence without overshooting the minimum of the loss function. | • Higher Learning Rate: Faster convergence but risk of overshooting, leading to suboptimal performance.<br>• Lower Learning Rate: Slower convergence with no significant accuracy improvement, increasing training time. |
| Batch size | 32 | Balances computational efficiency and model generalization. | • Larger Batch Size: Faster training but higher risk of overfitting.<br>• Smaller Batch Size: Improves generalization but increases training time and computational overhead. |
| Number of epochs | 50 | Determined by convergence behavior; early stopping applied if validation loss did not improve over 10 epochs. | • Increasing Epochs: Can improve model performance if underfitting but risks overfitting if not controlled.<br>• Decreasing Epochs: May result in underfitting if the model doesn't learn adequately within fewer epochs. |
| Dropout rate | 0.5 | Used to prevent overfitting by randomly dropping units during training, improving generalization. | • Lower Dropout Rate: Reduces overfitting control, increasing the risk of the model memorizing training data.<br>• Higher Dropout Rate: Excessive regularization can lead to underfitting by not learning enough features. |
| Optimization algorithm | Adam | Selected for its adaptive learning rate and momentum, facilitating faster convergence and handling sparse gradients. | • Switching to SGD: Slower convergence, potentially leading to longer training times and less efficient learning.<br>• Using Other Optimizers: Could impact the model's ability to reach the global minimum, depending on the optimizer's efficiency. |

and AUC-ROC scores. It's worth mentioning that the adoption of CLAHE has also increased sensitivity and specificity, which shows that the models have a better identification rate.

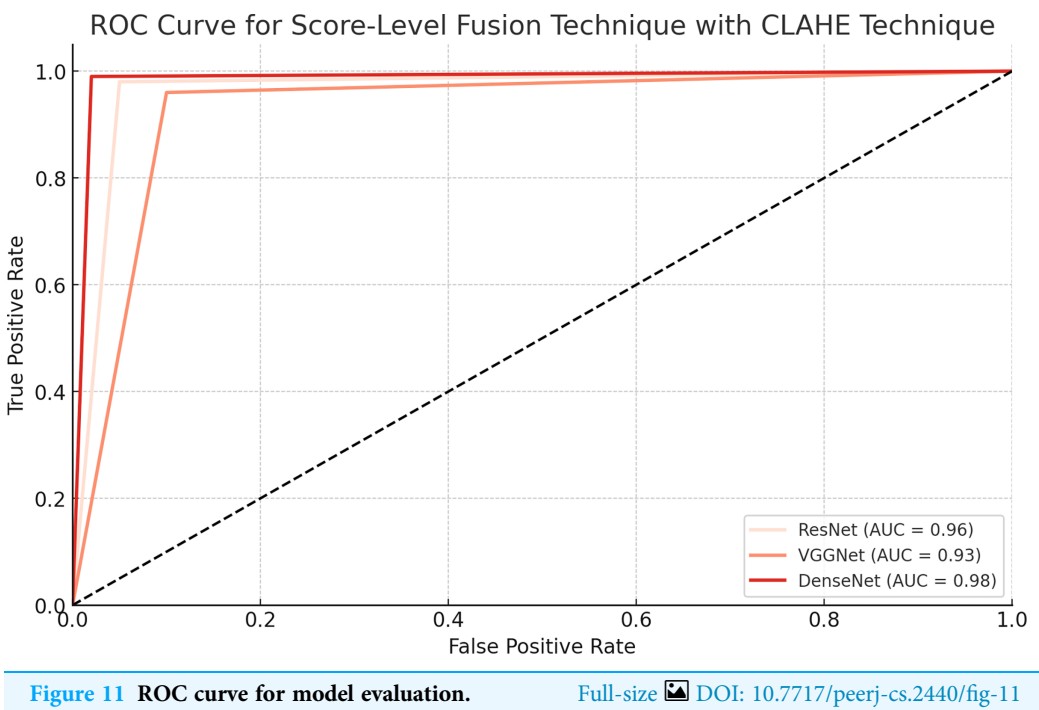

**Figure 11 ROC curve for model evaluation.**

- ResNet: A significant improvement in model performance was shown by the fact that ResNet's accuracy increased from 0.85 to 0.96 and its EER decreased from 10.5% to 5.0% when trained using CLAHE. Additionally, there was an improvement from 0.9 to 0.98 in the AUC-ROC, which stands for the tradeoff between specificity and sensitivity.

- VGGNet: Likewise, VGGNet exhibited noteworthy advancements as well. There was a decrease in EER from 12.3% to 6.8% and an improvement in accuracy from 0.83 to 0.93. The model's identification performance was significantly enhanced, as the AUC-ROC went from 0.88 to 0.96.

- DenseNet: Out of the three models, DenseNet showed the most performance. From 0.86 to 0.97 the accuracy had been increased, and from 9.8 to 4.5%, EER has been decreased. There was a significant improvement in the sensitivity/specificity ratio, as the AUC-ROC rose from 0.91 to 0.99.

To demonstrate the efficacy of CNN architectures and image enhancement methods like CLAHE, the NUPT-FPV dataset is used. This dataset contains images of both fingerprints and finger veins. There are two distinct physiological features that may be used for the identification of fingerprints and images of the veins of the fingers. Nevertheless, there are a number of variables that might impact the quality of these images during capture, which include sensor noise, pressure fluctuations, skin conditions, and brightness variations. Fortunately, CLAHE can help with a few of these problems, leading to better identification results. It is well known that the CLAHE method improves the efficiency of CNNs and other machine learning models that process image data by enhancing the images' contrast. The following are some of the ways in which the CLAHE method has

improved the identification accuracy rate of the CNN architectures employed in the research:

- Enhanced image details: By performing the normalization of the pixel intensity values throughout the image using histogram equalization methods, CLAHE increases the contrast of an image. To avoid noise amplification, a 'Contrast Limiting' approach is implemented. In applications like biometric identification, this augmentation may be vital since it brings to clear vision of certain features which were not invisible in the original image.

- Improved feature extraction: CNNs use pooling operations and convolutional filters to extract features from images; these processes depend significantly on the contrast between various areas of the image. Better feature extraction and, by augmentation, improved identification can be achieved by employing the CLAHE's enhanced contrast methods.

- Reduced impact of lighting conditions: Variations in lighting conditions can significantly impact the performance of image-based identification systems. By normalizing the intensity distribution throughout the image, CLAHE minimises these discrepancies and makes the CNN model more resilient to varying brightness conditions.

- Lower false positives and false negatives: The equalization performed by CLAHE had helped to reduce both false positives and false negatives. This is evident in the reduced EER, which is a measure of the point where both false positive and false negative rates are equal.

- Improved discrimination ability: The higher AUC-ROC values observed with the application of CLAHE indicate that the technique improves the models' ability to differentiate between classes. This is particularly crucial for identification tasks, since the model must be able to differentiate between several individuals.

## LIMITATIONS OF THE WORK

1. Dataset Limitations:
   - Large but not extensive: The NUPT-FPV dataset, despite being large and extensive, may fail to cover all possible variations of fingerprint and finger vein images as they can take in real world. Thus, the scope may not guarantee high generalization.
   - Challenge of diversity: It may originate from limited diversity of such aspects as age or ethnicity and the environment in terms of the equipment settings or lighting. The unexpected occurrences may be experienced in reality.

2. Model complexity:
   - High computational density: Deep CNN architectures including ResNet, VGGNet, and DenseNet, the model described herein, are computationally expensive due to training and inference.
   - Training time: The necessity to train a deep model along with multiple fusion imposes the risks of model overfitting. At the same time, integrating one specification or change takes too much time for rapid commercial deployment.

3. Fusion techniques:
- Implementation variability: The implementation of a particular fusion type, such as early, late, or score-level, may be different. At the same time, the performance may heavily rely on correct weight specification and validation. Thus, they can be quite sensitive.
- Integration complexity: Combining multiple biometric modalities and fusion techniques increases the complexity of the system, potentially leading to integration and maintenance challenges.

4. Preprocessing limitations:
- Enhanced artifacts: While CLAHE has an array of benefts including improved contrast and features visibility, it should not be the only solution. In fact, over-enhancement and under-enhancement create artifacts. Thus, it may not be suitable for each type of the biometric images.
- Parameter dependency: Parameter choices may differ for every biometric image.

We acknowledge that while our proposed method has shown promising results, there are always opportunities for further improvement and optimization. Based on the study's findings and limitations, the following modifications could be made: Using techniques like data augmentation (*e.g.*, rotation, scaling, color jittering) and incorporating synthetic data generation methods (*e.g.*, generative adversarial networks-GANs) can help create a more diverse dataset that covers a wider range of scenarios. This would improve the model's ability to generalize to real-world condition. Utilizing methods such as crowdsourcing and deploying mobile data collection apps can help gather real-world fingerprint and finger vein images from diverse populations and environments, ensuring the dataset reflects practical variability. Implementing transfer learning with pre-trained models or using knowledge distillation to transfer the learned features from larger models to smaller, more efficient ones could significantly reduce training time and computational resources. Methods like genetic algorithms or grid search optimization can be employed to find the optimal weights for fusion strategies, reducing sensitivity and improving robustness. Additionally, ensemble learning techniques, such as stacking and bagging, could provide more reliable fusion outcomes. Implementing machine learning-based adaptive preprocessing techniques, such as Adaptive Histogram Equalization (AHE) or using neural networks for image enhancement, can tailor the preprocessing to the specific characteristics of each biometric image, reducing the likelihood of artifacts.

## CONCLUSION AND FUTURE WORK

This research used the NUPT-FPV dataset, which contains images of fingerprints and finger veins, to evaluate the performance of the CNN models employed to differentiate between the individuals based on the biometric features. We discovered that the employed models' recognition accuracy was much improved when we used the CLAHE method for image enhancement. Our findings demonstrated that the CLAHE method improved feature extraction efficiency by reducing local contrast and increasing the uniform distribution of brightness and contrast. Consequently, the model performed better on a

number of performance metrics, including accuracy, AUC-ROC, sensitivity, and specificity. Consistently outperforming the other CNN designs, DenseNet performance is good when images are passed through CLAHE before giving input to the model. With DenseNet and Score-Level Fusion in particular, the CLAHE method has improved the performance of all three fusion algorithms across all architectures. The effectiveness of integrating state-of-the-art image fusion and enhancement methods to enhance CNN performance is shown in this article. This result demonstrates the significance of combining powerful deep learning models with excellent image enhancement methods to successfully address the intricacies of biometric identification jobs. Although our study yielded promising findings, there are other areas that might be investigated in further studies:

1. **Incorporating diverse datasets**: Our study was confined to the NUPT-FPV dataset. Including additional datasets with varying characteristics could further substantiate the robustness and adaptability of the approach we have used.

2. **Examining additional image enhancement techniques:** This study utilized CLAHE as an image enhancement technique. Future work could explore and evaluate other image enhancement methods such as Gabor filters, wavelet transforms, and others, to gain a more nuanced understanding of how image quality impacts identification accuracy.

3. **Exploring Other Fusion Techniques**: We evaluated early, late, and score-level fusion in our study. It would be beneficial to assess other fusion strategies to understand their potential impact on the performance of identification tasks.

4. **Using Different Machine Learning Models**: Besides CNNs, other machine learning architectures such as recurrent neural networks (RNNs), long short-term memory (LSTM) networks, and Transformer networks could be investigated for their effectiveness in dealing with such tasks.

5. **Investigating other biometric modalities**: Future research can extend to other biometric characteristics such as iris patterns, facial recognition, and gait analysis to gauge their relative advantages and challenges in biometric identification.

6. **Testing on real-time systems**: While our current study focused on evaluating the method using a controlled dataset to ensure comprehensive analysis and validation, we agree that implementing and testing it in a real-time system is an important next step. This would further demonstrate the practicality and robustness of our approach in real-world applications. We plan to explore real-time implementation as part of our future work to extend the scope of our research.

### Funding
This work was supported by the Deanship of Research and Graduate Studies at King Khalid University under grant number RGP2/462/45, the Princess Nourah bint Abdulrahman University Researchers Supporting Project under grant number

PNURSP2024R507 at Princess Nourah bint Abdulrahman University, Riyadh, Saudi Arabia, the Research Supporting Project at King Saud University, Riyadh, Saudi Arabia under grant number RSPD2024R787, and the Deanship of Scientific Research at Northern Border University, Arar, KSA under project number NBU-FFR-2024-451-10. The funders had no role in study design, data collection and analysis, decision to publish, or preparation of the manuscript.

### Grant Disclosures

The following grant information was disclosed by the authors:
King Khalid University: RGP2/462/45.
Princess Nourah bint Abdulrahman University, Riyadh, Saudi Arabia: PNURSP2024R507.
King Saud University, Riyadh, Saudi Arabia: RSPD2024R787.
Northern Border University, Arar, KSA: NBU-FFR-2024-451-10.

### Competing Interests

The authors declare that they have no competing interests.

### Author Contributions

- Amal Alshardan performed the experiments, performed the computation work, authored or reviewed drafts of the article, and approved the final draft.
- Arun Kumar conceived and designed the experiments, analyzed the data, prepared figures and/or tables, and approved the final draft.
- Mohammed Alghamdi performed the experiments, analyzed the data, authored or reviewed drafts of the article, and approved the final draft.
- Mashael Maashi conceived and designed the experiments, performed the computation work, prepared figures and/or tables, and approved the final draft.
- Saad Alahmari performed the experiments, analyzed the data, authored or reviewed drafts of the article, and approved the final draft.
- Abeer A. K. Alharbi conceived and designed the experiments, analyzed the data, authored or reviewed drafts of the article, and approved the final draft.
- Wafa Almukadi performed the experiments, performed the computation work, prepared figures and/or tables, and approved the final draft.
- Yazeed Alzahrani conceived and designed the experiments, performed the computation work, prepared figures and/or tables, and approved the final draft.

### Data Availability

The Python codes for our newly developed CNN-based deep learning architecture, incorporating fusion techniques (early fusion, late fusion, score-level fusion) and Contrast Limited Adaptive Histogram Equalization (CLAHE), are available in the Supplemental File.

The finger vein and fingerprint images used in this study are from the NUPT-FPV dataset at GitHub: https://github.com/REN382333467/NUPT-FPV.

## Supplemental Information

Supplemental information for this article can be found online at http://dx.doi.org/10.7717/peerj-cs.2440#supplemental-information.

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
