# Peer review of "Multimodal biometric identification: leveraging convolutional neural network (CNN) architectures and fusion techniques with fingerprint and finger vein data"

_PeerJ Computer Science, doi:10.7717/peerj-cs.2440_

## Round 0.1 · original submission · Major Revisions

Based on the referee reports, I recommend a major revision of the manuscript. The author should improve the manuscript, taking carefully into account the comments of the reviewers in the reports and resubmit the paper.

Reviewer 1 ·

Basic reporting

- The summary section of the study is not well organized. The steps of the proposed method should be briefly presented sequentially. In its current form, the proposed approach is not clearly understood.
- The manuscript requires proofreading.
- When used for the first time, abbreviations should be expanded, e.g. CLAHE in the introduction. All text should be checked for abbreviations.
- In the introduction part of the article, the contributions of the study should be given in detail and in a way that attracts the readers' attention (maybe in the form of items).
- There has been a lot of recent work on multi-modal biometric identification. However, when the sources in the related studies section of the article were examined, it was seen that this section was not up-to-date. Related work should be rewritten by analyzing the relevant literature, and information about current studies should also be included.
- Care should be taken to write variables in italics within the article.
- Likewise, care should be taken to write variables in italics in Figures and Algorithms.
- Figure 1 has low resolution.
- Figure 4, where the architecture of the proposed method is presented, is quite complex, I think it should be rearranged.
- The error in the cover text of Table 5 should be corrected.
- It would be more appropriate to give the information in Table 7 at the beginning of the Results section.

Experimental design

- More information should be given about the NUPT-FPV dataset and sample images should be given.
- Can the proposed method be tried with different deep-learning architectures? Why were ResNet, VGGNet, and DenseNet chosen?
- How was the training-test ratio of the dataset determined for the experimental results?

Validity of the findings

- The proposed method can be tested on a real-time system.
- Considering the limits of the study, what modifications can be made to the method?

Additional comments

- I examined the study titled "Multimodal Biometric Identification: Leveraging CNN Architectures and Fusion Techniques with Fingerprint and Finger Vein Data" in detail. The article needs to be re-evaluated according to the corrections I have written below.

Reviewer 2 ·

Basic reporting

All comments have been added in detail to the last section.

Experimental design

All comments have been added in detail to the last section.

Validity of the findings

All comments have been added in detail to the last section.

Additional comments

Review Report for PeerJ Computer Science
(Multimodal Biometric Identification: Leveraging CNN Architectures and Fusion Techniques with Fingerprint and Finger Vein Data)

1. Within the scope of the study, a dataset containing finger vein and fingerprint images was subjected to various data preprocessing processes, and biometric identification processes were performed with various deep learning models and fusion techniques.

2. In the introduction section, biometric systems and security, as well as the content of the study, were mentioned at a basic level.

3. In the related works section, similar studies in the literature were mentioned. However, in this section, in order to emphasize the importance of the subject more clearly and to express the difference and originality of this study from other studies in the literature, it is strongly recommended to add a literature comparison table consisting of columns such as "dataset, methods, data preprocessing, data augmentation, github, results, positive aspects, negative aspects". After this process, the differences and originality of this study from the literature should be stated more clearly in bullet points.

4. It was stated that NUPT-FPV was used as the dataset in the study. The reason for preferring this dataset, the status of other open source datasets in the literature and the reason for not preferring them, and also why we remained dependent on a single dataset should be explained in detail.

5. Before using the dataset, passing it through various data preprocessing steps such as CLAHE has increased the quality of the study. It is positive that the dataset is passed through various data proprocessing processes instead of using it raw.

6. The use and preference of fusion techniques such as early fusion and late fusion are sufficient. However, it is stated in this section that VGG, DenseNet and ResNet are preferred as deep learning models. Although there are many different and state-of-the-art deep learning models that can be used in the literature to solve this problem, it should be explained more clearly why these models are preferred. Have different experiments been conducted other than these models, has the positive/negative effect of the increase in the number of different models been examined, please explain.

7. There are deficiencies in the evaluation metrics section. ROC curves related to the AUC scores obtained should be added. In addition, although the details given in Table-7 are positive and sufficient, in this section, it should be explained in more detail how the traning parameter selections are determined and what the effect of the change in these values will be.

In conclusion, although the study is at a certain level in terms of both subject and originality, attention should definitely be paid to all the sections listed above in order to better contribute to the literature.

---

## Round 0.2 · accepted · Accept

Author has addressed reviewer comments properly. Thus I recommend publication of the manuscript.

Reviewer 1 ·

Basic reporting

No comment

Experimental design

No comment

Validity of the findings

No comment

Additional comments

The authors have made revisions taking into account the comments. I believe the publication is acceptable in its current form.

Reviewer 2 ·

Basic reporting

All comments have been added in detail to the last section.

Experimental design

All comments have been added in detail to the last section.

Validity of the findings

All comments have been added in detail to the last section.

Additional comments

Review Report for PeerJ Computer Science
(Multimodal Biometric Identification: Leveraging CNN Architectures and Fusion Techniques with Fingerprint and Finger Vein Data)

The reviewer comments were responded to in detail and sufficiently. I recommend that the paper be accepted.